# BLACK BOX FEATURE SELECTION WITH ADDITIONAL MUTUAL INFORMATION

## ABSTRACT

Many times, working with data starts with writing code to build a model from the features $\mathbf{x}$ to the response $\mathbf{y}$. Answering questions about the data can require understanding what parts of the features $\mathbf{x}$ influence the response $\mathbf{y}$. We show that using the KL-divergence within a randomization test discovers important features, while allowing for the reuse of model code. We call this method AMI-CRT. AMI-CRT requires running models for each randomization. We develop a faster variant called FAST-AMI-CRT, and show it is robust to errors in the randomization. Questions of feature importance can also be asked at the level of an individual sample. We provide an example to show that optimal predictive models are insufficient for instance-wise feature selection. We show that the estimators from FAST-AMI-CRT can also be reused to find important features in a particular instance. We evaluate our method on several simulation experiments, on a genomic dataset, a clinical dataset for hospital readmission, and on a subset of classes in ImageNet. Our methods outperform several baselines in various simulated datasets, identifies biologically significant genes, selects the most important predictors of a hospital readmission event, and identifies distinguishing image regions in an image-classification task.

## 1 INTRODUCTION

Model interpretation techniques aim to select features important for a response by reducing models (sometimes locally) to be human interpretable. However, the phrase model interpretation can be a bit of a misnomer. Any interpretation of a model must be imbued to the model by the population distribution that provides the data to train the model. In this sense, interpreting a model should be viewed as understanding the population distribution of data through the lens of a model. Existing methods for understanding the population distributions only work with particular models fit to the population, particular choices of test statistic, or particular auxiliary models for interpretation (Ribeiro et al., 2016; Lundberg and Lee, 2017). Such structural restrictions limit the applicability of these methods to a smaller class of population distributions. To be able to work in a black-box manner, feature selection methods can use models but must not require a particular structure in models used in selection processes.

Understanding the population distribution can be phrased as assessing whether a response is independent of a feature given the rest of the features; this test is called a conditional randomization test (Candes et al., 2018). Conditional randomization tests require test statistics. Test statistics like linear model coefficients (Barber et al., 2015) or correlation may miss dependence between the response and outcome. To avoid missing relationships between variables, we develop the notion of a *proper test statistic*. Proper test statistics are those whose power increases to one as the amount of data increases. Conditional independence implies the conditional-joint factorizes into conditional-marginals. Measuring the divergence between these distributions yields a proper test statistic. Of the class of integral probability metrics (Müller, 1997) and $f$-divergences (Csiszár, 1964), the KL-divergence simplifies estimation and allows for reuse of the model structures and code from the standard task of predicting the response from the features. Using the KL-divergence in this context has a natural interpretation; it is a measure of the additional information each feature provides about the outcome over the rest. This measure of information is known as the additional mutual information (AMI) (Ranganath and Perotte, 2018).

Our proposed procedure is called the additional mutual information conditional randomization test (AMI-CRT). AMI-CRT uses regressions to simulate data from the null for each feature and compares the additional mutual information (AMI) of the original data to the AMI of the simulations from

the null to assess significance. AMI-CRT works with any regression with probabilistic outputs like cross-entropy-trained neural networks. Training many regressions for each sample from the null can require substantial computation. While this is an embarrassingly parallel computation, we develop FAST-AMI-CRT that only requires a single model trained from null data. FAST-AMI-CRT uses an average of the model from the null with the model from the original data. We show this mixture guards against both variance in model training and poor estimation of the null. Though simple, AMI-CRT outperforms popular procedures for feature importance on a wide variety of simulated data, hospital records, and biological data.

Working with data sometimes requires interpreting individual datapoints. For example, a doctor may benefit from knowing which features for a particular patient relate to their health. The process of identifying features at a datapoint-level is called *instance-wise* feature selection (Ribeiro et al., 2016; Lundberg and Lee, 2017; Gimenez and Zou, 2019). We identify an issue in instance-wise feature selection, where even features selected using the true population distribution do not yield the features that were used to generate the response of an instance. The crux of this disparity is that the response generation process, conditional on the features, may use randomness to select features. We provide an example to demonstrate where instance-wise feature selection can go awry. We develop sufficient conditions for instance-wise feature selection to avoid this issue. The same regression estimates from AMI-CRT can be used to estimate feature-importances with minimal computational overhead, resulting in a method we term additional mutual information instance-wise feature selection (AMI-IW). We demonstrate AMI-IW on multiple simulations and image data. Across all of these tasks AMI-IW outperforms popular baselines.

**Related Work.** Permutation tests (Fisher, 1937) provide a test for marginal independence between each feature and the outcome. However, they fail to test conditional independence, which is required when covariates are dependent on each other. To address this, solutions like Sure Independence Screening (Barut et al., 2016; Fan and Lv, 2008) and Conditional Randomization Tests (Barber et al., 2015; Candes et al., 2018) have been proposed. These outline frameworks for conditional independence testing. However, they often make linearity or additive noise assumptions about the data generating distribution. Furthermore, they require the choice of a test statistic to capture some notion of conditional independence. The user of such frameworks is often burdened with the task of choosing this test statistic, which may require strong assumptions about the data generating distribution. Extending this approach to neural networks, Lu et al. (2018) propose a fully connected network whose weights are used as a test statistic. Though moving beyond linear models, their method is specific to fully connected networks. Tansey et al. (2018) propose holdout randomization tests (HRTs) that use empirical loss as a test-statistic. The loss they use for continuous-valued distributions of response is the mean-squared-error (MSE), which may ignore higher order dependencies between the response and features. Using AMI inside an HRT would capture these higher order dependencies. In our results, we adapt our test-statistic AMI to HRTs and show that this produces better calibration than other choices of loss. HRTs provide computational speed-ups over CRTs. However this speedup comes at the cost of robustness to poor estimations of the null feature distribution. We demonstrate empirically that FAST-AMI-CRT is robust to such poor estimations.

Beyond understanding the population distribution, some tasks require interpreting a population distribution on the level of an individual datapoint. Methods that test for conditional independence work under distributional notions of feature selection, but are not designed to identify the relevant features for a particular sample. To address this issue of "instance-wise feature selection," several methods have been proposed, including local perturbations (Simonyan et al., 2013; Sundararajan et al., 2017; Ribeiro et al., 2016) and fitting simpler auxiliary models to explain the predictions of a large model (Chen et al., 2018; Lundberg and Lee, 2017; Yoon et al., 2019; Turner, 2016; Štrumbelj and Kononenko, 2014; Shrikumar et al., 2017). Our instance-wise work is most similar to that of Burns et al. (2019), who repurpose the HRT framework to perform instance-wise feature selection, or Gimenez and Zou (2019), who define a conditional randomization test (CRT) procedure for subsets of the feature space. In general, however, the conditions under which instance-wise feature selection with predictive models may be possible are not well developed. We address this issue by first identifying a set of sufficient conditions under which instance-wise feature selection is always possible. We then show how estimators used in AMI-CRT can be repurposed for use in an instance-wise setting, yielding a procedure called the AMI-IW.

## 2 PROPER TESTS FOR FEATURE SELECTION

Practitioners of machine learning use feature selection to identify important features for their predictive task. One way to filter out important features is to find those that improve predictions given the rest of the features. This can be formalized through conditional independence. Let $\mathbf{x}_j$ be the $j$th feature of $\mathbf{x}$ and let $\mathbf{x}_{-j}$ be all features but the $j$th one. The goal is to discover a set $\mathcal{S}$ such that $\forall \mathbf{x}_j \notin \mathcal{S}, \mathbf{x}_j \perp \mathbf{y} \mid \mathbf{x}_{-j}$, where independence is with respect to the true population distribution $q$. The only knowledge about $q$ comes from a finite set of samples $\mathcal{D}_N := \{(\boldsymbol{x}^{(i)}, \boldsymbol{y}^{(i)})\}_{i=1}^N$ sampled from the population. This means that it is impossible to assess exact conditional independence. Therefore, in the finite sample setting, we must formulate a statistical hypothesis test.

A conditional randomization test (CRT) (Candes et al., 2018) defines a hypothesis test for conditional independence. For the $j$th feature, CRTs first compute a test statistic $t$ using the $N$ samples of data $\mathcal{D}_N$. CRTs place this statistic in a null distribution where samples of the $j$th feature $\mathbf{x}_j$ are replaced by samples of $\widetilde{\mathbf{x}}_j \mid \mathbf{x}_{-j}$ which by construction satisfy $\widetilde{\mathbf{x}}_j \perp \mathbf{y} \mid \mathbf{x}_{-j}$. Letting $\widetilde{\mathcal{D}}_{j,N}$ be a dataset where $\{\boldsymbol{x}_j^{(i)}\}_{i=1}^N$ has been replaced by $\{\widetilde{\boldsymbol{x}}_j^{(i)}\}_{i=1}^N$, the $p$-value for a CRT is

$$p_j(\mathcal{D}_N) = \underset{\substack{\forall i=1...N \\ \widetilde{\boldsymbol{x}}_j^{(i)} \sim q(\mathbf{x}_j | \mathbf{x}_{-j} = \boldsymbol{x}_{-j}^{(i)})}}{\mathbb{E}} \left[ \mathbb{1}\left( t(\mathcal{D}_N) \leq t(\widetilde{\mathcal{D}}_{j,N}) \right) \right] = \underset{\substack{\forall i=1...N \\ \widetilde{\boldsymbol{x}}_j^{(i)} \sim q(\mathbf{x}_j | \mathbf{x}_{-j} = \boldsymbol{x}_{-j}^{(i)})}}{\mathbb{E}} \left[ \mathbb{1}\left( t(\mathcal{D}_N) - t(\widetilde{\mathcal{D}}_{j,N}) \leq 0 \right) \right],$$
(1)

Under smoothness constraints, the $p$-value is uniform under the null because it computes the cumulative distribution function of the test statistic under the null. While CRTs provide a general method for conditional independence testing, they leave several components including the choice of test statistic unspecified.

### 2.1 CHOOSING THE RIGHT TEST STATISTIC

Imagine a test statistic $t(\cdot) = t(\{\boldsymbol{x}_j^{(i)}, \boldsymbol{y}^{(i)}\}_{i=1}^N)$ that uses only a feature $\mathbf{x}_j$ and the outcome $\mathbf{y}$. Any $p$-values computed using this test statistic would be meaningless when testing for conditional independence, as $t$ never considers the remaining features $\mathbf{x}_{-j}$. Therefore, particular choices for test statistics limit what can be tested. To address this, we introduce the concept of a *proper* test statistic.

**Definition 1.** *Proper Test Statistic: A test statistic $t(\mathcal{D}_N)$ is proper if p-values produced by the statistic converge to 0 when the null must be rejected, and are uniformly distributed otherwise. Using $t$ in Equation (1), this is:*

$$p_j(\mathcal{D}_N) \xrightarrow[N \to \infty]{d} \begin{cases} Uniform(0,1) & if \ \mathbf{x}_j \perp \mathbf{y} \mid \mathbf{x}_{-j} \\ 0 \ with \ probability \ 1 & if \ \mathbf{x}_j \not\perp \mathbf{y} \mid \mathbf{x}_{-j} \end{cases},$$
(2)

*where $\xrightarrow{d}$ indicates a convergence in distribution. Under the alternate hypothesis, which in the case of feature selection is $\mathbf{x}_j \not\perp \mathbf{y} \mid \mathbf{x}_{-j}$, the power to reject the null hypothesis must be 1, implying $p_j \to 0$. A proper test statistic requires that Equation (2) must hold for all distributions of $\mathbf{y}, \mathbf{x}$.*

Proper tests statistics in a CRT select the features in $\mathcal{S}$ as the data grows. Definition 1 mirrors the concept of a scoring rule (Gneiting and Raftery, 2007), which measures the calibration of a probabilistic prediction by a model. A *proper* scoring rule is one such that the highest expected score is obtained by a model that uses the true probability distribution to make predictions.

**Divergences are proper test statistics.** Conditional independence means the conditional distribution $r$ factorizes:

$$r(\mathbf{x}_j, \mathbf{y} \mid \mathbf{x}_{-j}) = r(\mathbf{x}_j \mid \mathbf{x}_{-j}) r(\mathbf{y} \mid \mathbf{x}_{-j}).$$
(3)

Divergences measure the closeness between two distributions. A divergence is zero when the two distributions are the same and positive otherwise. Computing any divergence $\mathcal{K}$, like an integral probability metric (Müller, 1997) or an $f$-divergence (Csiszár, 1964), between the left hand side and right hand side of Equation (3) would be a proper test statistic. Let $\mathcal{K}(a,b) \geq 0$ with equality holding only when $a$ is equal in distribution to $b$, then a proper test statistic $K_j(r) := \mathbb{E}_{r(\mathbf{x}_{-j})}[\mathcal{K}(r(\mathbf{x}_j, \mathbf{y} \mid \mathbf{x}_{-j}), r(\mathbf{x}_j \mid \mathbf{x}_{-j})r(\mathbf{y} \mid \mathbf{x}_{-j}))]$. A consistent estimator of this quantity is a proper test statistic (see

Appendix B.1). Casting conditional independence testing as divergence estimation reduces this test to fitting univariate regressions that can reuse pre-developed model code from the features to the response.

Define the resampling distribution $\tilde{q}_j = \tilde{q}_j(\widetilde{\mathbf{x}}_j \mid \mathbf{x}_{-j})q(\mathbf{y}, \mathbf{x}_{-j})$. Using a divergence in a CRT requires estimates of the following conditional distributions: $q(\mathbf{x}_j \mid \mathbf{x}_{-j}), q(\mathbf{y} \mid \mathbf{x}), \tilde{q}_j(\mathbf{y} \mid \widetilde{\mathbf{x}}_j, \mathbf{x}_{-j})$, and $q(\mathbf{y} \mid \mathbf{x}_{-j})$. The first distribution $q(\mathbf{x}_j \mid \mathbf{x}_{-j})$ is required for any CRT. The next distribution $q(\mathbf{y} \mid \mathbf{x})$ corresponds to the standard task of building a good regression model. The third distribution $\tilde{q}_j(\mathbf{y} \mid \widetilde{\mathbf{x}}_j, \mathbf{x}_{-j})$ requires a regression model with corrupted inputs. This regression can reuse the model structure and code from the standard regression task $q(\mathbf{y} \mid \mathbf{x})$. However, the last distribution $q(\mathbf{y} \mid \mathbf{x}_{-j})$ could require development of new model structures. For example, if $\mathbf{x}$ is an image, a good model for $q(\mathbf{y} \mid \mathbf{x})$ could be a convolutional neural network. If the conditioning set $\mathbf{x}_{-j}$ is a subregion of that image, the convolutional neural network used for $q(\mathbf{y} \mid \mathbf{x})$ would need to be modified for different padding and filter sizes. This means new models could be needed for each $\mathbf{x}_{-j}$. In the next section, we show that the KL-divergence removes the need for estimating this distribution, and therefore only requires the piece needed for all CRTs, $q(\mathbf{x}_j \mid \mathbf{x}_{-j})$, and model code to fit the response from the features.

## 2.2 AMI-CRT

Using the KL-divergence as a test statistic in Equation (1) requires the difference $\delta_j$:

$$
\begin{aligned}
\delta_j =& \mathbb{E}_{q(\mathbf{x}_{-j})}[D_{\mathrm{KL}}(q(\mathbf{x}_j, \mathbf{y} \mid \mathbf{x}_{-j}), q(\mathbf{x}_j \mid \mathbf{x}_{-j})q(\mathbf{y} \mid \mathbf{x}_{-j}))] \\
& - \mathbb{E}_{\tilde{q}_j(\mathbf{x}_{-j})}[D_{\mathrm{KL}}(\tilde{q}_j(\widetilde{\mathbf{x}}_j, \mathbf{y} \mid \mathbf{x}_{-j}), \tilde{q}_j(\widetilde{\mathbf{x}}_j \mid \mathbf{x}_{-j})\tilde{q}_j(\mathbf{y} \mid \mathbf{x}_{-j}))] \\
=& \mathbb{E}_{q(\mathbf{x}_{-j})}[\mathbb{E}_{q(\mathbf{x}_j, \mathbf{y}\mid\mathbf{x}_{-j})}[\log q(\mathbf{x}_j, \mathbf{y} \mid \mathbf{x}_{-j}) - \log(q(\mathbf{x}_j \mid \mathbf{x}_{-j})q(\mathbf{y} \mid \mathbf{x}_{-j}))] \\
& - \mathbb{E}_{\tilde{q}_j(\widetilde{\mathbf{x}}_j, \mathbf{y}\mid\mathbf{x}_{-j})}[\log \tilde{q}_j(\widetilde{\mathbf{x}}_j, \mathbf{y} \mid \mathbf{x}_{-j}) - \log(\tilde{q}_j(\widetilde{\mathbf{x}}_j \mid \mathbf{x}_{-j})\tilde{q}_j(\mathbf{y} \mid \mathbf{x}_{-j}))]] \\
=& \mathbb{E}_{q(\mathbf{x}_{-j})}[\mathbb{E}_{q(\mathbf{x}_j, \mathbf{y}\mid\mathbf{x}_{-j})}[\log q(\mathbf{y} \mid \mathbf{x}) - \log q(\mathbf{y} \mid \mathbf{x}_{-j})] \\
& - \mathbb{E}_{\tilde{q}_j(\widetilde{\mathbf{x}}_j, \mathbf{y}\mid\mathbf{x}_{-j})}[\log \tilde{q}_j(\mathbf{y} \mid \mathbf{x}_{-j}) - \log \tilde{q}_j(\mathbf{y} \mid \mathbf{x}_{-j})]] \\
=& \mathbb{E}_{q(\mathbf{x}_{-j})}[\mathbb{E}_{q(\mathbf{x}_j, \mathbf{y}\mid\mathbf{x}_{-j})}[\log q(\mathbf{y} \mid \mathbf{x})] - \mathbb{E}_{\tilde{q}_j(\widetilde{\mathbf{x}}_j, \mathbf{y}\mid\mathbf{x}_{-j})}[\log \tilde{q}_j(\mathbf{y} \mid \widetilde{\mathbf{x}}_j, \mathbf{x}_{-j})]] \\
=& \mathbb{E}_{q(\mathbf{x}, \mathbf{y})}[\log q(\mathbf{y} \mid \mathbf{x})] - \mathbb{E}_{q(\mathbf{x}_{-j}, \mathbf{y})}\mathbb{E}_{\tilde{q}_j(\widetilde{\mathbf{x}}_j\mid\mathbf{x}_{-j})}[\log \tilde{q}_j(\mathbf{y} \mid \widetilde{\mathbf{x}}_j, \mathbf{x}_{-j})]].
\end{aligned}
$$

The second equality above follows from $q(\mathbf{x}_{-j}) = \tilde{q}_j(\mathbf{x}_{-j})$, and the fourth from $q(\mathbf{y} \mid \mathbf{x}_{-j}) = \tilde{q}_j(\mathbf{y} \mid \mathbf{x}_{-j})$. This simplification of $\delta_j$ means that for the computation of the average KL-divergence as a test statistic, the distribution $q(\mathbf{y} \mid \mathbf{x}_{-j})$ is unecessary, thereby reducing computation and allowing the reuse of training infrastructure for predicting the response $\mathbf{y}$. The KL-divergence provides this reduction since $\log$ splits products into sums. This expected KL-divergence is called the additional mutual information (AMI) (Ranganath and Perotte, 2018).

Computing the expected value of $\delta_j$ to get the $p$-value in Equation (1) requires estimation from a finite sample. Recall that $\mathcal{D}_N$ is a collection of $N$ datapoints sampled iid from $q(\mathbf{x}, \mathbf{y})$. These samples from the population come from the provided data. Similarly, $\widetilde{\mathcal{D}}_{j,N}$ is sampled iid from $q(\mathbf{x}_{-j}, \mathbf{y})q(\widetilde{\mathbf{x}}_j \mid \mathbf{x}_{-j})$. The samples $\mathbf{x}_{-j}, \mathbf{y}$ come from the population distribution, however sampling $q(\widetilde{\mathbf{x}}_j \mid \mathbf{x}_{-j})$ requires learning a model for this distribution also known as the complete conditional. The other two pieces of $\delta_j$ require building a model from $\mathbf{x}$ to $\mathbf{y}$ and $(\widetilde{\mathbf{x}}_j, \mathbf{x}_{-j})$ to $\mathbf{y}$. To learn these models, the data $\mathcal{D}_N$ is split into training and test sets. Let $\theta, \beta$ be parameters, $q_\theta(\widetilde{\mathbf{x}}_j \mid \mathbf{x}_{-j})$ and $q_\beta(\mathbf{y} \mid \mathbf{x})$ are fit to the training set. The distribution $q_\gamma(\mathbf{y} \mid \widetilde{\mathbf{x}}_j, \mathbf{x}_{-j})$ is fit to the training part of $\widetilde{\mathcal{D}}_{j,N}$. To compute the expectations in $\delta_j$, these models are evaluated on their corresponding test sets. Finally, a Monte Carlo estimate of the expectation for the $p$-value in Equation (1) requires replicating this procedure $K$ times. We call this procedure AMI-CRT (Algorithm 1).

The models $q_\theta(\widetilde{\mathbf{x}}_j \mid \mathbf{x}_{-j})$ and $q_\beta(\mathbf{y} \mid \mathbf{x})$ can be shared across the replications, however $q_\gamma(\mathbf{y} \mid \widetilde{\mathbf{x}}_j, \mathbf{x}_{-j})$ must be recomputed for each replication. While this is an embarrassingly parallel problem, it involves the computation of a new model for each draw from the null distribution. To speed up the computation of AMI-CRT, we use the average of the original model $q_\beta(\mathbf{y} \mid \mathbf{x})$ and a single model trained for $q_\gamma(\mathbf{y} \mid \widetilde{\mathbf{x}}_j, \mathbf{x}_{-j})$. This averaged model is used to estimate the two terms in the AMI-difference $\delta_j$. Under the null, the estimates are identically distributed because each estimate evaluates the same function on identically distributed samples. Thus, averaging produces $p$-values that are uniform under the null, but the average may not result in a proper test-statistic. However,

the averaged model performs almost as well as AMI-CRT empirically. We call this procedure the FAST-AMI-CRT, which is summarized in Algorithm 2.

Averaging is needed because models trained on data drawn from the same distribution have variance (Friedman et al., 2001). The averaged model provides FAST-AMI-CRT with several advantages. First, it is more conservative than using just the original model as in HRTs (Tansey et al., 2018) since the averaged model both predicts better on the null data and worse on the real data. We show empirically that this guards against errors in the estimation of the complete conditional distribution. Second, it requires only a single null model per feature instead of one per replication.

To estimate each of $q_\theta$ and $q_\beta$, standard regression models like logistic regression, neural networks, and random forests can be used at no more computational cost than training. Nonparametric regression can be used as well. The choice between these estimators should be made by using the best fitting regression on validation data. The estimation procedure is straightforward, yet effective as we demonstrate in Section 4. In the next section, we show how the building blocks for FAST-AMI-CRT can be used to provide feature importances on an instance-wise level.

## 3    TESTS FOR INSTANCE-WISE FEATURE SELECTION

So far, we show how to recover features important across the whole population. We have not yet addressed the issue that different samples could have different important features. We call this problem of recovering important features for each sample, *instance-wise feature selection (*IWFS*)*. To identify important features instance-wise, we can use the probability of observing a particular label $\boldsymbol{y}^{(i)}$ given a set of features $\boldsymbol{x}^{(i)}$. This suggests a candidate definition for important features:

**Definition 2.** *(Candidate) Feature importance in* IWFS*: Let $q$ be the true population distribution. Let $\{(\boldsymbol{x}^{(i)}, \boldsymbol{y}^{(i)})\}_{i=1}^N$ be a dataset where each $(\boldsymbol{x}^{(i)}, \boldsymbol{y}^{(i)}) \sim q$ . The $j$th feature for the $i$th sample, $\boldsymbol{x}_j^{(i)}$ , is an important feature if*
$$q(\mathbf{y} = \boldsymbol{y}^{(i)} \mid \mathbf{x} = \boldsymbol{x}^{(i)}) > q(\mathbf{y} = \boldsymbol{y}^{(i)} \mid \mathbf{x}_{-j} = \boldsymbol{x}_{-j}^{(i)}).$$

Definition 2 says that a feature $\boldsymbol{x}_j^{(i)}$ is important if observing it increases the probability of $\boldsymbol{y}^{(i)}$. This formulation is exploited in (Yoon et al., 2019; Chen et al., 2018) to obtain instance-wise important features. However, Definition 2 can sometimes fail to identify relevant features, even with access to the true conditional distribution $q(\mathbf{y} \mid \mathbf{x})$. While important features may satisfy this condition, so will a few unimportant features. As a demonstrative example, consider the data generating process where $\mathbf{y} = \mathbf{z}\mathbf{x}_1 + (1-\mathbf{z})\mathbf{x}_2 + \epsilon$, $\mathbf{z} \sim \text{Bernoulli}(0.5)$, $\mathbf{x}_1, \mathbf{x}_2 \sim \mathcal{N}(0, \sigma_\mathbf{x}^2)$, and $\epsilon \sim \mathcal{N}(0, \sigma_\epsilon^2)$. Assume we have the true $q(\mathbf{y} \mid \mathbf{x}_1, \mathbf{x}_2)$, and let $\mathbf{z}$ be unobserved. Pick any sample $(\boldsymbol{x}_1^{(i)}, \boldsymbol{x}_2^{(i)}, \boldsymbol{y}^{(i)})$ where the corresponding $\boldsymbol{z}^{(i)} = 1$, meaning that $\boldsymbol{x}_1^{(i)}$ is important for this instance. We can expand:

$$q(\boldsymbol{y}^{(i)}|\boldsymbol{x}_1^{(i)}, \boldsymbol{x}_2^{(i)}) - q(\boldsymbol{y}^{(i)}|\boldsymbol{x}_1^{(i)}) = \frac{1}{2}\mathcal{N}\left(\boldsymbol{y}^{(i)}; \boldsymbol{x}_2^{(i)}, \sigma_\epsilon^2\right) - \frac{1}{2}\mathcal{N}\left(\boldsymbol{y}^{(i)}; 0, \sigma_\boldsymbol{x}^2 + \sigma_\epsilon^2\right)$$

For all $i$ such that $\boldsymbol{y}^{(i)}$ lies in a non-0 interval around $\boldsymbol{x}_2^{(i)}$, we see that $q(\boldsymbol{y}^{(i)}|\boldsymbol{x}_1^{(i)}, \boldsymbol{x}_2^{(i)}) - q(\boldsymbol{y}^{(i)}|\boldsymbol{x}_1^{(i)}) > 0$. For example if $\sigma_\epsilon^2 = \sigma_\mathbf{x}^2 = 1$, and $\boldsymbol{x}_2^{(i)} = 5$, then $\boldsymbol{y}^{(i)} \in [3, 7]$ will violate this inequality. In all of those cases, the wrong feature will be selected as important as per the candidate Definition 2. We show the full derivation of this example in Appendix C.1. The fundamental issue with the formulation in Definition 2 is that noise can act as a "selection" mechanism, but cannot be estimated because it is unobserved. While predictive models $q_\beta$ for $q(\mathbf{y} \mid \mathbf{x})$ suffice for understanding population distributions, they might not be sufficient to perform IWFS.

### 3.1    SUFFICIENT CONDITIONS FOR INSTANCE-WISE FEATURE SELECTION

We develop the following condition under which $q(\mathbf{y} \mid \mathbf{x})$ is sufficient to perform IWFS:

**Proposition 1.** *Sufficient conditions for instance-wise feature selection:    For each sample $(\boldsymbol{x}^{(i)}, \boldsymbol{y}^{(i)})$, let $\mathcal{S}^{(i)}$ be a set of features that contribute to the prediction of $\boldsymbol{y}^{(i)}$ defined as:*

$$\mathcal{S}^{(i)} := \left\{\boldsymbol{x}_j^{(i)} : q(\mathbf{y} = \boldsymbol{y}^{(i)} \mid \mathbf{x} = \boldsymbol{x}^{(i)}) > q(\mathbf{y} = \boldsymbol{y}^{(i)} \mid \mathbf{x}_{-j} = \boldsymbol{x}_{-j}^{(i)})\right\}. \tag{4}$$

*If $\mathbf{y}$ is discrete, and $q(\mathbf{y} = \boldsymbol{y}^{(i)} \mid \mathbf{x} = \boldsymbol{x}^{(i)}) = 1$ for each sample $(\boldsymbol{x}^{(i)}, \boldsymbol{y}^{(i)})$, i.e. we have perfect predictions on our dataset, then it is possible to recover such a set $\mathcal{S}^{(i)}$ for all $i$.*

| Feature | ami-crt | fast-ami-crt | loss-hrt | corr-crt | lime | shap | rf |
|---|:---:|:---:|:---:|:---:|:---:|:---:|:---:|
| Provides $p$-values | ✓ | ✓ | ✓ | ✓ | | | |
| Well calibrated $p$-values | ✓ | ✓ | | | | | |
| Instance-wise feature selection | ✓ | ✓ | ✓ | | ✓ | ✓ | |
| No distributional assumptions | ✓ | ✓ | ✓ | | ✓ | ✓ | ✓ |
| Robust to $q(\mathbf{x}_j \mid \mathbf{x}_{-j})$ estimation | ✓ | ✓ | | | | | |

**Table 1:** Both AMI-CRT and FAST-AMI-CRT produce well-calibrated $p$-values, provide false discovery rate (FDR) control (Benjamini and Hochberg, 1995), allow instance-wise feature selection, and make no distributional assumptions about the data-generating process. This table compares these methods to widely-used feature selection methods.

The set in Equation (4) consists of only features $\boldsymbol{x}_j^{(i)}$ that help increase the likelihood of observing $\boldsymbol{y}^{(i)}$ given the remaining features $\boldsymbol{x}_{-j}^{(i)}$. If the perfect predictions property of $q(\mathbf{y} \mid \mathbf{x})$ is true, then $q(\mathbf{y} = \boldsymbol{y}^{(i)} \mid \mathbf{x}_{-j} = \boldsymbol{x}_{-j}^{(i)})$ can only be less than or equal to $q(\mathbf{y} = \boldsymbol{y}^{(i)} \mid \mathbf{x} = \boldsymbol{x}^{(i)})$, with equality when $\boldsymbol{x}_j^{(i)}$ is not important to $\boldsymbol{y}^{(i)}$. Assuming the sufficient conditions in Proposition 1, we can now construct an IWFS procedure using the same estimators from AMI-CRT or FAST-AMI-CRT.

### 3.2 AMI INSTANCE-WISE FEATURE SELECTION

Instance-wise feature selection can be performed using the building blocks of the FAST-AMI-CRT. Starting from Definition 2, we begin by manipulating $q(\mathbf{y} = \boldsymbol{y}^{(i)} \mid \mathbf{x} = \boldsymbol{x}^{(i)})$ and marginalizing out $\boldsymbol{x}_j^{(i)}$. We then use Jensen's inequality to upper-bound the log of this expectation as follows:

$$\mathbb{E}_{\widetilde{\boldsymbol{x}}_j^{(i)} \sim q(\mathbf{x}_j \mid \boldsymbol{x}_{-j}^{(i)})} \left[ -\log \tilde{q}_j(\mathbf{y} = \boldsymbol{y}^{(i)} \mid \mathbf{x}_j = \widetilde{\boldsymbol{x}}_j, \mathbf{x}_{-j} = \boldsymbol{x}_{-j}^{(i)}) \right] \geq -\log q(\mathbf{y} = \boldsymbol{y}^{(i)} \mid \mathbf{x}_{-j} = \boldsymbol{x}_{-j}^{(i)}) \tag{5}$$

This suggests the following instance-wise test. If the inequality in Equation (5) is strict, the feature is considered important. If equality holds in Equation (5), the feature is considered unimportant. Notice that Jensen's inequality could introduce slack in this bound that could make a feature seem relevant when it is not. We use Proposition 1 to show that this is not an issue.

Recall that given a model $q_\beta$ for $q(\mathbf{y} \mid \mathbf{x})$ that satisfies the instance-wise sufficient conditions in Proposition 1, $q_\beta(\mathbf{y} = \boldsymbol{y}^{(i)} \mid \mathbf{x} = \boldsymbol{x}^{(i)}) \geq q_\beta(\mathbf{y} = \boldsymbol{y}^{(i)} \mid \mathbf{x}_{-j} = \boldsymbol{x}_{-j}^{(i)})$. In the case where $\boldsymbol{x}_j^{(i)}$ does not help predict $\boldsymbol{y}^{(i)}$, $\tilde{q}_j(\mathbf{y} = \boldsymbol{y}^{(i)} \mid \mathbf{x}_j = \widetilde{\boldsymbol{x}}_j^{(i)}, \mathbf{x}_{-j} = \boldsymbol{x}_{-j}^{(i)})$ cannot be greater than $q(\mathbf{y} = \boldsymbol{y}^{(i)} \mid \mathbf{x}_{-j} = \boldsymbol{x}_{-j}^{(i)})$, as the former does not depend on $\widetilde{\boldsymbol{x}}_j^{(i)}$. Then the left-hand side of Equation (5) becomes $-\log q(\mathbf{y} = \boldsymbol{y}^{(i)} \mid \mathbf{x}_{-j} = \boldsymbol{x}_{-j}^{(i)})$, a constant with respect to $\widetilde{\boldsymbol{x}}_j^{(i)}$, implying an equality. Therefore, checking for equality in Equation (5) is a valid test to see if a feature is either important or unimportant. In Appendix C.2, we detail an example that shows how scores computed using Equation (5) can help rank features from most to least helpful for prediction. We term this procedure the additional mutual information instance-wise feature selection (AMI-IW).

If we computed an expectation over $\boldsymbol{x}^{(i)}, \boldsymbol{y}^{(i)}$ of Equation (5), this procedure resembles FAST-AMI-CRT. We can reuse the estimators from FAST-AMI-CRT to compute these instance-wise log-probability differences for AMI-IW. Therefore, we only use one null estimator for $\tilde{q}_j(\mathbf{y} \mid \widetilde{\mathbf{x}}_j, \mathbf{x}_{-j})$. Like FAST-AMI-CRT, we can potentially use a mixture of estimators in AMI-IW, but at the cost of power to select important features.

## 4 EXPERIMENTS

We compare our methods, the AMI-CRT [ami-crt] and fast additional mutual information conditional randomization test (FAST-AMI-CRT) [fast-ami-crt] to widely-used approaches on various performance metrics. The baselines are:

| Dataset | ami-crt | fast-ami-crt | loss-hrt | corr-crt | lime | shap | rf |
|---|---|---|---|---|---|---|---|
| orange | **0.97** | 0.95 | 0.94 | 0.22 | 0.94 | 0.95 | 0.94 |
| xor | **1.00** | 0.97 | 0.95 | 0.45 | **1.00** | 0.99 | 0.95 |

**Table 2:** Simulated data results: Here we use the scores provided by each method to select features. AMI-CRT's area under the receiver operating characteristic (ROC) curve is better than other of state-of-the-art methods.

| | Dataset | ami-iw | loss-hrt | corr-crt | lime | shap | rf |
|---|---|---|---|---|---|---|---|
| Precision | selector | **0.93** | 0.78 | 0.34 | 0.58 | 0.64 | 0.33 |
| | noisy-selector | **0.67** | 0.45 | 0.33 | 0.57 | 0.61 | 0.33 |
| Selector identification | selector | **0.95** | 0.88 | 0.34 | 0.35 | 0.39 | 0.33 |
| | noisy-selector | **0.97** | 0.85 | 0.33 | 0.25 | 0.37 | 0.33 |

**Table 3:** Instance-wise feature selection results. The precision experiment measures the ability of each selection method to identify relevant features while restricted to 7 features. The selector identification experiment counts the instances where each method identified the selector feature $\mathbf{x}_1$ (relevant for all instances).

- Correlation [corr-crt]: Difference between Correlation$(\mathbf{x}_j, \mathbf{y})$ and Correlation$(\widetilde{\mathbf{x}}_j, \mathbf{y})$ as a test statistic for a CRT
- Local interpretable model-agnostic explanations (LIME) [lime] (Ribeiro et al., 2016)
- Shapley additive explanations (SHAP) [shap] (Lundberg and Lee, 2017)

- Random forest [rf] feature importance scores
- Zero-one loss [loss-hrt]: Binary classification loss of $q_\beta(\mathbf{y} \mid \mathbf{x})$ where $q_\beta$ is a model for $\mathbf{y} \mid \mathbf{x}$ is used as a test statistic in a HRT (Tansey et al., 2018)

HRTs construct a test by comparing a loss function evaluated on $\mathbf{x}$ and $\widetilde{\mathbf{x}}_j, \mathbf{x}_{-j}$. The choice of loss, however, is left to the practitioner. We study the 0-1 loss, which is a proper scoring rule, in all of our experiments. In specific settings, we equip an HRT with our AMI test-statistic. This method is called the `ami-hrt`. We show that `ami-hrt` is better calibrated than 0-1 HRT. Table 1 presents a summary comparison of the properties of each selection method. We use the regression approach using conditional categorical distribution parameterized with neural networks highlighted in (Miscouridou et al., 2018) to model $q(\mathbf{x}_j|\mathbf{x}_{-j})$ for all experiments unless specified otherwise.

**Simulations:** We simulate data for evaluating each selection method. These tests are designed to highlight the differences between each method.

[xor]: To test the case where features on their own are not informative, but together provide information, we use the `xor` dataset. We first sample $\mathbf{x} \sim \mathcal{N}(0, \Sigma_D)$ $N$ times, where $\Sigma_D$ is a $D$-dimensional covariance matrix. We translate the first two dimensions of each sample $\boldsymbol{x}^{(i)}$ away from the origin in 4 different directions : $\{(s,s), (-s,s), (s,-s), (-s,-s)\}$ with uniform probability. If the resulting translation has first two coordinates with the same sign, the label is one. Otherwise, it is zero. All but the first two features are independent of $\mathbf{y}$. We set $N = 2000, D = 20$.

[orange](Chen et al., 2018): To test the case where $\mathbf{y}$ is some nonlinear function of $\mathbf{x}$, we use the `orange` dataset. In this dataset, $\mathbf{x} \sim \mathcal{N}(0, \Sigma_D)$, $\mathbf{y} = 1$ if $\exp\left(\sum_{j=1}^{\ell} \mathbf{x}_j^2 - \ell\right) > 0.5$ and 0 otherwise, where $\ell < D$ is the number of important features. We choose $N = 3000, D = 20$, and $\ell = 4$.

[selector, noisy-selector]: These experiments test instance-wise feature selection methods. We first sample $\mathbf{x} \sim \mathcal{N}(0, \Sigma_D)$ $N$ times, where $\Sigma_D$ is a $D$-dimensional covariance matrix, and $D \geq 11$. The first feature $\mathbf{x}_1$, called the "selector" feature, determines the feature selection mechanism. We generate $\mathbf{y} \in \{0, 1\}$ as Equation (6). We also investigate the effectiveness of each feature selection method in the presence of noise, and generate $\mathbf{y} \in \{0, 1\}$ as Equation (7). We set the parameter $N = 2000, D = 20$.

$$q(\mathbf{y} = 1 \mid \mathbf{x}) = \begin{cases} \langle \beta_1, \mathbf{x}_{2:6} \rangle > 0 & \text{if } \mathbf{x}_1 > 0 \\ \langle \beta_2, \mathbf{x}_{7:11} \rangle > 0 & \text{if } \mathbf{x}_1 \leq 0 \end{cases} \quad (6) \qquad q(\mathbf{y} = 1 \mid \mathbf{x}) = \begin{cases} \sigma(\langle \beta_1, \mathbf{x}_{2:6} \rangle) & \text{if } \mathbf{x}_1 > 0 \\ \sigma(\langle \beta_2, \mathbf{x}_{7:11} \rangle) & \text{if } \mathbf{x}_1 \leq 0 \end{cases} \quad (7)$$

***Results.*** For methods based on CRTs or HRTs, we select features using $p$-values. For the baselines that do not produce $p$-values, we select features using the importance scores provided by each method. We threshold $p$-values or importance scores respectively, and compute an ROC curve for each method. We present the mean area under each curve over 100 simulations for the xor and orange datasets in Table 2. We notice that this task is easily solved by most methods apart from corr-crt. This test fails to account for dependencies between features. The ami-crt achieves a higher AUROC than baselines, while fast-ami-crt achieves similar performance.

To identify important features in practice, a threshold for importance scores must be chosen. If a method produces $p$-values, we can control the false discovery rate (FDR) (Benjamini and Hochberg, 1995). This is the expected proportion of falsely identified features. An assumption for standard FDR-controlling procedures is independent $p$-values (Benjamini and Hochberg, 1995). Therefore, we investigate the calibration of $p$-values across ami-crt, fast-ami-crt, corr-crt, and loss-hrt. We omit other baselines in this comparison as they do not produce $p$-values and therefore have no direct FDR-control.

To evaluate each method, we use the generating process for the orange dataset, and set $N = 3000, D = 104, \ell = 4$. If the $p$-values are independent, null $p$-values should resemble iid draws from a Uniform(0,1) distribution. Figure 3 (Appendix D) shows a quantile-quantile plot of null $p$-values. We also perform a Kolmogorov-Smirnov (KS) (Massey Jr, 1951) test where the null distribution is Uniform(0,1). All CRT-based methods produce independent $p$-values, while loss-hrt produces deflated and significantly non-uniform $p$-values ($p = 0.0006$), implying dependence. As a result loss-hrt incorrectly identifies many null features as important. Models for each replication of null features ami-crt potentially decrease the correlation of the $p$-values. The fast-ami-crt achieves a middle ground between the hrt and ami-crt by yielding well-calibrated $p$-values while requiring only one null model $q_\gamma(\mathbf{y} \mid \widetilde{\mathbf{x}}_j, \mathbf{x}_{-j})$ per feature. We also investigate the use of ami-hrt, which uses the AMI as a test statistic in an HRT in Figure 3.

To better understand the difference between refitting (CRTs) and not refitting (HRTs), we inspect the robustness of each method to poor simulations from the null (poor estimation of $q(\mathbf{x}_j \mid \mathbf{x}_{-j})$). We use the orange dataset and set all off-diagonal values of $\Sigma_D$ to 0.5. We sample $\widetilde{\mathbf{x}}_j$ from $\mathcal{N}(0, 1)$. We see in Figure 4 (Appendix D) that refitting protects against poor approximations of $q(\mathbf{x}_j \mid \mathbf{x}_{-j})$; the $p$-value distribution for fast-ami-crt is uniform, while the $p$-value distribution for both HRT methods is significantly non-uniform ($p = 5 \times 10^{-6}$ for ami-hrt and $p = 10^{-7}$ for loss-hrt). The robustness of fast-ami-crt comes from averaging which makes predictions for the original data worse and predictions on the null better regardless of the quality of the null simulations.

For *instance-wise feature selection*, we perform two tests for each selection method. To test HRTs in this setting, we use the procedure prescribed by Burns et al. (2019) but with different test statistics. For precision, we identify the 7 (the true # of relevant features per instance) most important features as dictated by each selection method and report average precision scores across a held-out test set in Table 3. For selector identification, we count the number of instances where the selector variable $\mathbf{x}_1$ was identified. For the selector task, we notice that the ami-iw achieves the highest precision, followed by the loss-hrt. In the noisy-selector task, we notice a decrease in scores across all methods with the largest decrease for ami-iw and loss-hrt. The noisy-selector case violates the sufficient conditions for instance-wise feature selection (Equation (4)) meaning that the noise in sampling the response can obscure which features are important. This explains the reduction in performance. Even with the decrease in performance, ami-iw performs best.

The ami-iw and loss-hrt identify the selector variable $\mathbf{x}_1$ in nearly every sample in our test set. Linear methods like lime fail because the selection mechanism is highly non-linear. Further, rf and corr-crt are not designed to assign importance at the level of an individual sample and therefore do not provide meaningful scores per instance.

**Wellcome Trust Celiac disease**: We study data from a genomic analysis on Celiac disease (Dubois et al., 2010). For each individual in this dataset, we have a set of single nucleotide polymorphisms (SNPs). SNPs represents genetic variance in the individual with respect to some reference population. This dataset consists of two classes of individuals: cases ($n = 3796$) and controls ($n = 8154$), where the cases are those with Celiac disease. After standard preprocessing steps as prescribed by Bush

and Moore (2012), 1759 SNPs remain. To model $q(\mathbf{x}_j|\mathbf{x}_{-j})$, we use the same procedures as (Candes et al., 2018) where we estimate $q(\mathbf{x}_j|\mathbf{x}_{S_j})$ where $S_j$ is only the set of SNPs (not including $\mathbf{x}_j$) known to be correlated with $\mathbf{x}_j$. To model $q(\mathbf{y}|\mathbf{x})$, we use an $L_1$-logistic regression model.

***Results.*** In Table 4, we show the results for all methods with FDR-control. We identify the SNPs that most likely contribute to distinguishing between those with Celiac disease and those without it. Since these methods produce $p$-values, we can select features at a theoretical FDR of 20% using the Benjamini and Hochberg (1995) procedure. We report the percentage of selected SNPs that have been previously shown to be associated with Celiac disease in a biological context as reported by one of (Dubois

| Method | # Selected | Precision | Recall |
|---|---|---|---|
| ami-crt | 17 | **76.47%** | **32.5%** |
| fast-ami-crt | 16 | 68.75% | 27.5% |
| loss-hrt | 14 | 57.14% | 20.0% |
| corr-crt | *185* | 6.40% | 30.0% |

**Table 4:** The number of significant features reported at a 20% FDR level for each test, and the percentage of features previously identified in a biological study.

et al., 2010; Sollid, 2002; Adamovic et al., 2008; Hunt et al., 2008). There are 40 SNPs in total that are both in our dataset and in these papers. Ami-crt outperforms all other methods tested; fast-ami-crt performs similarly. We also list the SNPs returned by ami-crt in Appendix E. As expected, corr-crt selects a large set of features, but achieves fairly low precision. This is because many SNPs are correlated with each other, and all of these seem relevant marginally. The AMI-based methods have better precision and recall compared to loss-hrt potentially both due to aforementioned deviation from uniform and that the zero-one loss may not change when only one out of more than a thousand features gets perturbed.

**Hospital readmission**: We use a dataset consisting of ten years of medical logs from over 130 hospitals (Strack et al., 2014). Features in the dataset include time spent in the hospital, medical specialty of attending doctor, age, and various other diagnostic information. Labels for each sample represent one of three events: readmitted within the next 30 days ($n = 35,545$), readmitted after 30 days ($n = 11,357$), or not readmitted ($n = 54,864$). Due to class imbalance, we grouped all readmitted patients into one category ($n = 46,902$). We detail further preprocessing steps in Appendix F. To model $q(\mathbf{y}|\mathbf{x})$, we use a random forest classifier with 100 estimators.

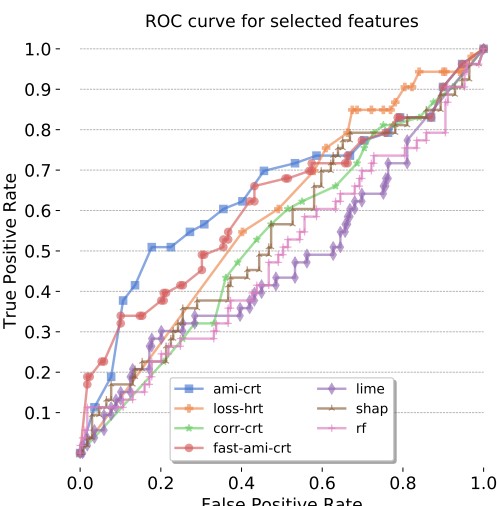

**Figure 1:** Using results from (Strack et al., 2014) as a clinically validated ground truth, we observe that AMI-CRT is able to achieve the highest area under the receiver operating characteristic curve (AUROC) when compared to state-of-the-art benchmarks.

***Results.*** The ground truth features come from clinical validation done by (Strack et al., 2014). We use importance scores (or $p$-values) estimated by each selection method with these ground truth features to compute an ROC curve. Figure 1 shows these curves for each method. We observe that ami-crt and fast-ami-crt achieve a higher area under the ROC curve than state-of-the-art approaches. The loss-hrt performs well, but achieves low power at false positive rates less than 0.5. These methods, unlike locally-linear methods such as lime and shap, do not assume that relevant features are marginally independent of irrelevant features (Lundberg and Lee, 2017).

**ImageNet experiments:** We consider the task of differentiating between ambulances and policevans. This task is interesting as both objects are physically very similar and there are only a few features that can be used to differentiate the two. For example, both objects have windows, wheels, and doors, so other features must be used to distinguish between the two classes. Rather than consider each pixel as an individual feature $\mathbf{x}_j$, we consider a patch of pixels $\mathbf{x}_S$ as a single feature, such that no two patches contain overlapping pixels. To model the distribution

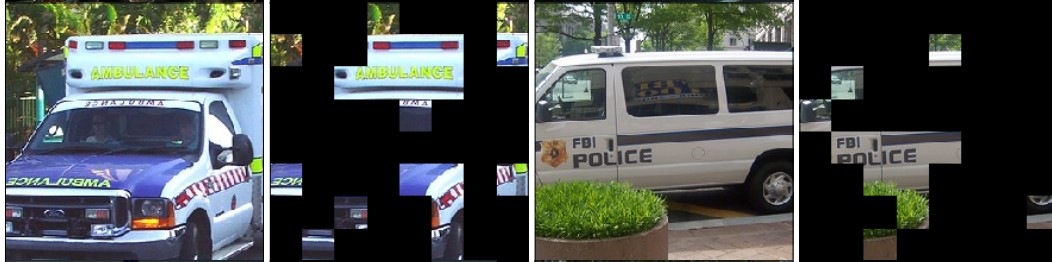

**Figure 2:** Instance-wise feature selection using AMI. The first and third columns show the original image of ambulances or policevans respectively. The second and fourth columns show only the patches which were found to have non-zero AMI with the label, given the rest of the patches.

$q(\mathbf{x}_S|\mathbf{x}_{-S})$, we make use of a generative inpainting model $\pi_g$ (Yu et al., 2018). We split the image into an $8 \times 8$ grid so that there are $64$ non-overlapping $\mathbf{x}_S$ patches. To model $q(\mathbf{y}|\mathbf{x})$, we use a VGG-16 network (Simonyan and Zisserman, 2015). To perform our instance-wise test, we compute log-probability differences using fifty generated samples from $q(\mathbf{x}_S|\mathbf{x}_{-S})$ for each patch.

***Results.*** In Figure 2, we show a subset of results of AMI-IW. The first and third columns show the original images for each class: ambulance and policevan respectively. The second and fourth columns mask out the original image in patches where the patch is not found to be relevant to the prediction. The model used to estimate $q(\mathbf{y}|\mathbf{x})$ is able to achieve roughly $90\%$ accuracy on a held-out test set. We see that our predictive model uses relevant details like the words "ambulance" or "police" printed on the vehicle to distinguish between each class. The model also tends to ignore objects like windscreens and other features shared across classes, as is expected. These results indicate that the difference in log probabilities between a model using the true data, and one using $\widetilde{\mathbf{x}}_S$ sampled from $q(\mathbf{x}_S \mid \mathbf{x}_{-S})$ works well in determining a relevant set of features even on an instance-wise level. We show several additional images in Figure 5, in Appendix G. We also compare our method to local interpretable model-agnostic explanations (LIME) and shapley additive explanations (SHAP) (Figures 6 and 7). Both methods perform reasonably well on this task, but identify objects that are known to be common to both classes like wheels and headlamps. Neither method identifies writing on the vehicles in the images. This is likely because of the simplifying assumptions made by these locally-linear methods. They assume that the set of relevant features is independent of the set of irrelevant features, which may not be the case in images. For example, the location of the word "ambulance" may depend on the window position.

## 5 DISCUSSION

We develop AMI-CRT for testing for conditional independence of each feature $\mathbf{x}_j \perp \mathbf{y} \mid \mathbf{x}_{-j}$ from a finite sample from the population distribution. AMI-CRT uses the KL-divergence to cast independence testing as regression and allows for the reuse of code from building the original model from the features to the response. We develop FAST-AMI-CRT which requires less computation than AMI-CRT and is robust to poor estimation of the null conditional. We define sufficient conditions under which to perform instance-wise feature selection and develop the AMI-IW, an instance-wise feature selection method built from the pieces of FAST-AMI-CRT. AMI-CRT, FAST-AMI-CRT, and AMI-IW all outperform several popular methods. in various simulated tasks, in identifying biologically significant genes, selecting the most indicative features to predict hospital readmissions, and in identifying distinguishing features in an image classification task.

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

# A  ALGORITHMS

## A.1  AMI-CRT ALGORITHM

We show the AMI-CRT procedure in Algorithm 1, and the FAST-AMI-CRT procedure in Algorithm 2.

---

**Algorithm 1:** $k$-fold AMI feature selection

---

**Input:** $\boldsymbol{x} \in \mathbb{R}^{N \times D}$, feature matrix; $\boldsymbol{y} \in \mathbb{R}^N$, labels
**Output:** $\mathbf{p}$, the $p$-values for $\boldsymbol{x}_j \perp \boldsymbol{y} \,|\, \boldsymbol{x}_{-j} \,\forall j$
Split $\boldsymbol{x}$ and $\boldsymbol{y}$ into $M$ folds, $G_1, G_2, \ldots, G_M$
**for** $j \in [1, 2, \ldots, D]$ **do**
    Estimate $q_\theta := q_\theta(\boldsymbol{x}_j \,|\, \boldsymbol{x}_{-j})$
    $h \leftarrow 0$
    **for** $m \in [1, 2, \ldots, M]$ **do**
        Estimate $q_\beta^{(m)} := q_\beta(\boldsymbol{y}^{(G_{-m})} \,|\, \boldsymbol{x}^{(G_{-m})})$
        $h \leftarrow h + \frac{1}{M}\mathcal{L}_\beta(\boldsymbol{y}^{(G_m)} \,|\, \boldsymbol{x}^{(G_m)})$, where $\mathcal{L}_\beta$ is an log-likelihood estimate using $q_\beta^{(m)}$
    **end**
    Let $\tilde{\mathbf{h}}$ be a $K$-dimensional vector of 0s
    **for** $k \in [1, 2, \ldots, K]$ **do**
        Sample $\tilde{\boldsymbol{x}}_j \sim q_\theta$
        **for** $m \in [1, 2, \ldots, M]$ **do**
            Estimate $q_{\tilde{\beta}}^{(k,m)} := q_{\tilde{\beta}}(\boldsymbol{y}^{(G_{-m})} \,|\, \tilde{\boldsymbol{x}}_j^{(G_{-m})}, \boldsymbol{x}_{-j}^{(G_{-m})})$
            Let $\tilde{\mathbf{h}}^{(k)} \leftarrow \tilde{\mathbf{h}}^{(k)} + \frac{1}{M}\mathcal{L}_{\tilde{\beta}}(\boldsymbol{y}^{(G_m)} \,|\, \tilde{\boldsymbol{x}}_j^{(G_m)}, \boldsymbol{x}_{-j}^{(G_m)})$, where $\mathcal{L}_{\tilde{\beta}}$ is an log-likelihood
             estimate using $q_{\tilde{\beta}}^{(k,m)}$
        **end**
    **end**
    Let $\mathbf{p}_j = \frac{1}{K+1}\left[1 + \sum_{k=1}^{K} \mathbb{1}\left(h \leq \tilde{\mathbf{h}}^{(k)}\right)\right]$
**end**

---

## A.2  FAST-AMI-CRT ALGORITHM

---

**Algorithm 2:** FAST-AMI-CRT

---

**Input:** $\boldsymbol{x} \in \mathbb{R}^{N \times D}$, feature matrix; $\boldsymbol{y} \in \mathbb{R}^N$, labels
**Output:** $\mathbf{p}$, the $p$-values for $\boldsymbol{x}_j \perp \boldsymbol{y} \,|\, \boldsymbol{x}_{-j} \,\forall j$
Fit $q_\beta := q_\beta(\boldsymbol{y} \,|\, \boldsymbol{x})$
**for** $j \in [1, 2, \ldots, D]$ **do**
    Fit $q_\theta := q_\theta(\boldsymbol{x}_j \,|\, \boldsymbol{x}_{-j})$
    Let $\tilde{\boldsymbol{x}}$ be a dataset such that $\tilde{\boldsymbol{x}}_{-j} = \boldsymbol{x}_{-j}$, and $\tilde{\boldsymbol{x}}_j$ is randomly sampled from $q_\theta(\boldsymbol{x}_j \,|\, \boldsymbol{x}_{-j})$
    Fit $q_\gamma := q_\beta(\boldsymbol{y} \,|\, \tilde{\boldsymbol{x}}_j, \boldsymbol{x}_{-j})$
    Let $h_j \leftarrow \mathbb{E}\left[\log\left(\frac{1}{2}q_\beta(\boldsymbol{y} \,|\, \boldsymbol{x}) + \frac{1}{2}q_\gamma(\boldsymbol{y} \,|\, \boldsymbol{x})\right)\right]$ with respect to dataset $(\boldsymbol{x}, \boldsymbol{y})$
    **for** $k \in [1, 2, \ldots, K]$ **do**
        Let $\tilde{\boldsymbol{x}}^{(k)}$ be a dataset such that $\tilde{\boldsymbol{x}}_{-j} = \boldsymbol{x}_{-j}$, and $\tilde{\boldsymbol{x}}_j$ is randomly sampled from
          $q_\theta(\boldsymbol{x}_j \,|\, \boldsymbol{x}_{-j})$
        Let $\tilde{h}_j^{(k)} \leftarrow \mathbb{E}\left[\log\left(\frac{1}{2}q_\beta(\boldsymbol{y} \,|\, \tilde{\boldsymbol{x}}) + \frac{1}{2}q_\gamma(\boldsymbol{y} \,|\, \tilde{\boldsymbol{x}})\right)\right]$ with respect to dataset $(\tilde{\boldsymbol{x}}, \boldsymbol{y})$
    **end**
    Let $\mathbf{p}_j = \frac{1}{K+1}\left[1 + \sum_{k=1}^{K} \mathbb{1}\left(h \leq \tilde{h}_j^{(k)}\right)\right]$
**end**

# B  PROOFS AND DERIVATIONS

## B.1  DIVERGENCES ARE PROPER TEST STATISTICS

In this section, we prove that divergences are proper test statistics. Let $\mathcal{D}_N := \{\boldsymbol{x}_j^{(i)}, \boldsymbol{y}^{(i)}, \boldsymbol{x}_{-j}^{(i)}\}_{i=1}^N$ and $\widetilde{\mathcal{D}}_{j,N} := \{\widetilde{\boldsymbol{x}}_j^{(i)}, \boldsymbol{y}^{(i)}, \boldsymbol{x}_{-j}^{(i)}\}_{i=1}^N$. We first list the assumptions here:

1. Let $t(\mathcal{D}_N)$ be a consistent estimator of $\mathbb{E}_{q(\mathbf{x}_{-j})}[\mathcal{K}(q(\mathbf{x}_j, \mathbf{y} \mid \mathbf{x}_{-j}), q(\mathbf{x}_j \mid \mathbf{x}_{-j}) q(\mathbf{y} \mid \mathbf{x}_{-j}))]$.

2. Let $t(\widetilde{\mathcal{D}}_{j,N})$ be a consistent estimator of $\mathbb{E}_{\tilde{q}_j(\mathbf{x}_{-j})}[\mathcal{K}(\tilde{q}_j(\widetilde{\mathbf{x}}_j, \mathbf{y} \mid \mathbf{x}_{-j}), \tilde{q}_j(\widetilde{\mathbf{x}}_j \mid \mathbf{x}_{-j}) \tilde{q}_j(\mathbf{y} \mid \mathbf{x}_{-j}))]$.

3. The cumulative distribution functions of $t(\mathcal{D}_N)$ and $t(\widetilde{\mathcal{D}}_{j,N})$ are both continuous everywhere.

4. We have access to complete conditionals $q(\widetilde{\mathbf{x}}_j \mid \mathbf{x}_{-j})$

*Proof.* We prove that $t$ is a *proper* test statistic if and *only if* $t(\mathcal{E}_n)$ is a consistent estimator of $\mathbb{E}_{r(\mathbf{x}_{-j})}[\mathcal{K}(r(\mathbf{x}_j, \mathbf{y} \mid \mathbf{x}_{-j}), r(\mathbf{x}_j \mid \mathbf{x}_{-j}) r(\mathbf{y} \mid \mathbf{x}_{-j}))]$. We do this by showing $t$ yields $p$-values that are zero under the alternate hypothesis and uniformly distributed under the null.

Recall that the $p$-value for our test is:

$$p_j(\mathcal{D}_N) = \mathbb{E}_{\widetilde{\boldsymbol{x}}_j^{(i)} \sim q(\boldsymbol{x}_j \mid \boldsymbol{x}_{-j}^{(i)})} \left[ \mathbb{1} \left( t(\mathcal{D}_N) - t(\widetilde{\mathcal{D}}_{j,N}) \le 0 \right) \right].$$

**Under the alternate hypothesis**    Consider the case where $\mathbf{x}_j \not\perp \mathbf{y} \mid \mathbf{x}_{-j}$. As $N \to \infty$,

$$t(\widetilde{\mathcal{D}}_{j,N}) \xrightarrow{q} \mathbb{E}_{\tilde{q}_j(\mathbf{x}_{-j})} \mathcal{K} \left( \tilde{q}_j(\widetilde{\mathbf{x}}_j, \mathbf{y} \mid \mathbf{x}_{-j}) \parallel \tilde{q}_j(\widetilde{\mathbf{x}}_j \mid \mathbf{x}_{-j}) \tilde{q}_j(\mathbf{y} \mid \mathbf{x}_{-j}) \right) = 0,$$

where $\xrightarrow{\tilde{q}_j}$ indicates a convergence in probability.

Since $\mathbf{x}_j \not\perp \mathbf{y} \mid \mathbf{x}_{-j}$, notice also that

$$t(\mathcal{D}_N) \xrightarrow{q} \mathbb{E}_{q(\mathbf{x}_{-j})} \mathcal{K} \left( q(\mathbf{x}_j, \mathbf{y} \mid \mathbf{x}_{-j}) \parallel q(\mathbf{x}_j \mid \mathbf{x}_{-j}) q(\mathbf{y} \mid \mathbf{x}_{-j}) \right) > 0,$$

Therefore, the term inside the expectation in the $p_j(\mathcal{D}_N)$ above is always 0, yielding a $p$-value of 0 in the limit of $N$. Since these $p$-values converge in probability to a single point, the $p$-values converge in distribution to a delta mass at 0.

**Under the null hypothesis**    In the case where $\mathbf{x}_j \perp \mathbf{y} \mid \mathbf{x}_{-j}$, the samples in $q_N(\mathbf{y}, \mathbf{x})$ and $\tilde{q}_{j,N}(\mathbf{y}, \widetilde{\mathbf{x}}_j, \mathbf{x}_{-j})$ are both sampled from the same distribution $q = \tilde{q}_j$. Therefore, the distribution of $t(\mathcal{D}_N)$ as a function of $q_N$, is the same as that of $t(\widetilde{\mathcal{D}}_{j,N})$ as a function of $\tilde{q}_{j,N}$.

Let $F_N$ be the cumulative distribution function of $t(\widetilde{\mathcal{D}}_{j,N})$ which in this case is the same as that of $t(\mathcal{D}_N)$.

We rewrite the $p$-value expression as $p_j^N := p_j(\mathcal{D}_N) = P\left( t(\widetilde{\mathcal{D}}_{j,N}) \le t(\mathcal{D}_N) \right) = F_N(t(\mathcal{D}_N))$. Now let $F_N^{-1}(\cdot)$ be the *generalized inverse cumulative distribution function* which exists because $F_N$ is a continuous everywhere function. With this, we derive the distribution of the $p$-value:

$$P(p_j^N \le \rho) = P(F_N^{-1}(p_j^N) \le F_N^{-1}(\rho)) = P(t(\mathcal{D}_N) \le F_N^{-1}(\rho)) = F_N(F_N^{-1}(\rho)) = \rho.$$

This means $p_j^N$ is uniformly distributed under the null. This result holds for all values of $N$. Thus $p_j^N$ forms a sequence of random variables, indexed by $N$, that are identically distributed as a uniform random variable over $[0, 1]$. This means that the sequence converges in distribution to a uniform distribution over $[0, 1]$.

This shows that $t$ is a proper test statistic.

$\square$

**Continuity of $F_N$**  Earlier, we assumed the cumulative distribution function (CDF) $F_N(t(\mathcal{D}_N)) = P(t(\widetilde{\mathcal{D}}_{j,N}) \leq t(\mathcal{D}_N))$ was continuous everywhere. This is required for the generalized inverse distribution function $F_N^{-1}$ to be well-defined on the full range $[0, 1]$.

Discontinuities could occur when the event $t(\mathcal{D}_N) = t(\widetilde{\mathcal{D}}_{j,N})$ occurs with some non-zero probability $c$. This means that the $p$-value does not take all the values in $[0, 1]$. To see this, note that $F_N(\cdot) \notin [x, x + c)$ for some $x \in [0, 1 - c)$.

To remedy this, we can replace the indicator function in our test-statistic with the following function:

$$\mathbf{1}(a, b) = \begin{cases} 1 & \text{if } a < b \\ 0 & \text{if } a > b \, , \\ \text{Uniform}([0, 1]) & \text{if } a = b \end{cases}$$

where $\text{Uniform}([0, 1])$ is a continuous uniform random variable.

With this new function, consider the events, $t(\mathcal{D}_N) = t(\widetilde{\mathcal{D}}_{j,N})$ and $t(\mathcal{D}_N) \neq t(\widetilde{\mathcal{D}}_{j,N})$. Conditioning on $t(\mathcal{D}_N) = t(\widetilde{\mathcal{D}}_{j,N})$, the distribution of the $p$-value is the same as the uniform random variable : $\text{Uniform}([0, 1])$. Also note that the distribution $F(\cdot \mid t(\mathcal{D}_N) \neq t(\widetilde{\mathcal{D}}_{j,N}))$ is continuous everywhere in its support because $t(\mathcal{D}_N) = t(\widetilde{\mathcal{D}}_{j,N})$ occurs with zero probability.

Thus, this modification ensures that the $p$-values are distributed uniformly.

## C  INSTANCE-WISE FEATURE SELECTION EXAMPLES

### C.1  CONSISTENT PREDICTIONS ALONE ARE INSUFFICIENT FOR INSTANCE-WISE FEATURE SELECTION

Recall our sufficiency condition for instance-wise feature selection as mentioned in Proposition 1. In this example, we see what happens when this condition is not met. We notice that this definition does not suffice to reject an unimportant feature. Consider a simple data generating process:

$$\mathbf{y} = \mathbf{z}\mathbf{x}_1 + (1 - \mathbf{z})\mathbf{x}_2 + \epsilon$$
$$\mathbf{z} \sim \text{Bernoulli}(0.5)$$
$$\mathbf{x}_1, \mathbf{x}_2 \sim \mathcal{N}(0, \sigma_\mathbf{x}^2)$$
$$\epsilon \sim \mathcal{N}(0, \sigma_\epsilon^2)$$

where $\mathbf{z}$ is not observed. We can now write out the probability distributions we care about. Note that taking an expectation like $\mathbb{E}_{\widetilde{\mathbf{x}}_1 \sim q(\mathbf{x}_1 | \mathbf{x}_2)} q(\mathbf{y} | \widetilde{\mathbf{x}}_1 \mathbf{x}_2)$ yields $q(\mathbf{y} | \mathbf{x}_2)$. For simplicity, we leave out the use of complete conditions and work directly with the latter probability distributions:

$$p(\mathbf{y} | \mathbf{x}_1, \mathbf{x}_2, \mathbf{z}) = \mathcal{N}\left(\mathbf{z}\mathbf{x}_1 + (1 - \mathbf{z})\mathbf{x}_2, \sigma_\epsilon^2\right)$$
$$p(\mathbf{y} | \mathbf{x}_1, \mathbf{x}_2) = \int p(\mathbf{y} | \mathbf{x}_1, \mathbf{x}_2, \mathbf{z}) p(\mathbf{z} | \mathbf{x}_1, \mathbf{x}_2) dz = \int p(\mathbf{y} | \mathbf{x}_1, \mathbf{x}_2, \mathbf{z}) p(\mathbf{z}) dz$$
$$= \frac{1}{2} \mathcal{N}\left(\mathbf{x}_1, \sigma_\epsilon^2\right) + \frac{1}{2} \mathcal{N}\left(\mathbf{x}_2, \sigma_\epsilon^2\right)$$
$$p(\mathbf{y} | \mathbf{x}_2) = \frac{1}{2} \mathcal{N}\left(0, \sigma_\mathbf{x}^2 + \sigma_\epsilon^2\right) + \frac{1}{2} \mathcal{N}\left(\mathbf{x}_2, \sigma_\epsilon^2\right)$$
$$p(\mathbf{y} | \mathbf{x}_1) = \frac{1}{2} \mathcal{N}\left(\mathbf{x}_1, \sigma_\epsilon^2\right) + \frac{1}{2} \mathcal{N}\left(0, \sigma_\mathbf{x}^2 + \sigma_\epsilon^2\right)$$

Now consider an instance $(\boldsymbol{x}_1^{(i)}, \boldsymbol{x}_2^{(i)}, \boldsymbol{y}^{(i)}, \boldsymbol{z}^{(i)})$ where $\boldsymbol{z}^{(i)} = 1$ which means that $\boldsymbol{y}^{(i)}$ depends only on feature $\boldsymbol{x}_1^{(i)}$. Now we check if $p(\boldsymbol{y}^{(i)} | \boldsymbol{x}_1^{(i)}, \boldsymbol{x}_2^{(i)}) > p(\boldsymbol{y}^{(i)} | \boldsymbol{x}_1^{(i)})$. Using our definitions from

earlier, we can expand this inequality:

$$p(\boldsymbol{y}^{(i)}|\boldsymbol{x}_1^{(i)}, \boldsymbol{x}_2^{(i)}) - p(\boldsymbol{y}^{(i)}|\boldsymbol{x}_1^{(i)}) = \frac{1}{2}\mathcal{N}\left(\boldsymbol{y}^{(i)}; \boldsymbol{x}_2^{(i)}, \sigma_\epsilon^2\right) - \frac{1}{2}\mathcal{N}\left(\boldsymbol{y}^{(i)}; 0, \sigma_{\boldsymbol{x}}^2 + \sigma_\epsilon^2\right)$$

For all $i$ such that $\boldsymbol{y}^{(i)}$ lies in a non-0 interval around $\boldsymbol{x}_2^{(i)}$, we have that $p(\boldsymbol{y}^{(i)}|\boldsymbol{x}_1^{(i)}, \boldsymbol{x}_2^{(i)}) - p(\boldsymbol{y}^{(i)}|\boldsymbol{x}_1^{(i)}) > 0$. For example let $\sigma_\epsilon = \sigma_x = 1$, then $\boldsymbol{x}_2^{(i)} = 5$, we have that $\boldsymbol{y} \in [3, 17]$ satisfies this. This means that $x_2$ will be deemed important as per the candidate Proposition 1.

### C.2 INSTANCE-WISE SCORE EXAMPLE

In this example, we see how scores computed using Equation (5) can help identify important features for a given instance, under the assumptions stated in Proposition 1. Consider a simple data generating process:

$$\mathbf{y} = \mathbf{z}\mathbf{x}_1 + (1 - \mathbf{z})\mathbf{x}_2$$
$$\mathbf{z}, \mathbf{x}_1, \mathbf{x}_2 \sim \text{Bernoulli}(0.5)$$

where all random variables are observed. Let us now consider the following observed instance: $(\boldsymbol{y}^{(i)}, \boldsymbol{x}_1^{(i)}, \boldsymbol{x}_2^{(i)}, \boldsymbol{z}^{(i)}) = (1, 0, 1, 0)$. We can now devise a test for each of $\boldsymbol{x}_1^{(i)}$, $\boldsymbol{x}_2^{(i)}$, and $\boldsymbol{z}^{(i)}$. For $\boldsymbol{x}_1^{(i)}$, we want to check:

$$p(\boldsymbol{y}^{(i)}|\boldsymbol{x}_1^{(i)}, \boldsymbol{x}_2^{(i)}, \boldsymbol{z}^{(i)}) > p(\boldsymbol{y}^{(i)}|\boldsymbol{x}_2^{(i)}, \boldsymbol{z}^{(i)}) \tag{8}$$

We can create similar tests for the other two variables as well:

$$p(\boldsymbol{y}^{(i)}|\boldsymbol{x}_1^{(i)}, \boldsymbol{x}_2^{(i)}, \boldsymbol{z}^{(i)}) > p(\boldsymbol{y}^{(i)}|\boldsymbol{x}_1^{(i)}, \boldsymbol{z}^{(i)}) \tag{9}$$
$$p(\boldsymbol{y}^{(i)}|\boldsymbol{x}_1^{(i)}, \boldsymbol{x}_2^{(i)}, \boldsymbol{z}^{(i)}) > p(\boldsymbol{y}^{(i)}|\boldsymbol{x}_1^{(i)}, \boldsymbol{x}_2^{(i)}) \tag{10}$$

We can use Table 5 to help evaluate Equations (8) to (10):

$$p(\boldsymbol{y}^{(i)} = 1|\boldsymbol{x}_1^{(i)} = 0, \boldsymbol{x}_2^{(i)} = 1, \boldsymbol{z}^{(i)} = 0) = 1$$
$$p(\boldsymbol{y}^{(i)} = 1|\boldsymbol{x}_2^{(i)} = 1, \boldsymbol{z}^{(i)} = 0) = 1$$
$$p(\boldsymbol{y}^{(i)} = 1|\boldsymbol{x}_1^{(i)} = 0, \boldsymbol{z}^{(i)} = 0) = 0.5$$
$$p(\boldsymbol{y}^{(i)} = 1|\boldsymbol{x}_1^{(i)} = 0, \boldsymbol{x}_2^{(i)} = 1) = 0.5$$

meaning $\boldsymbol{x}_1^{(i)}$ is not important to $\boldsymbol{y}^{(i)}$, but $\boldsymbol{x}_2^{(i)}$ and $\boldsymbol{z}^{(i)}$ are important.

| $\boldsymbol{x}_1$ | $\boldsymbol{x}_2$ | $\boldsymbol{z}$ | $\boldsymbol{y}$ |
|---|---|---|---|
| 0 | 0 | 0 | 0 |
| 0 | 0 | 1 | 0 |
| 0 | 1 | 0 | 1 |
| 0 | 1 | 1 | 0 |
| 1 | 0 | 0 | 0 |
| 1 | 0 | 1 | 1 |
| 1 | 1 | 0 | 1 |
| 1 | 1 | 1 | 1 |

**Table 5:** Full distribution for example in Appendix C.2

## D SIMULATED DATA FEATURE SELECTION - ADDITIONAL RESULTS

In this section, we present additional results that use AMI as a test statistic in a HRT framework. This offers a significant speedup as the HRT framework avoids having to refit estimators using $x_j \sim q(\mathbf{x}_j \mid \mathbf{x}_{-j})$. Figure 3 shows a quantile-quantile plot of the null $p$-values for each FDR-controlling feature selection method. We notice that both HRT-based methods tend to deflate $p$-values. This often results in features being mistakenly selected as important. Using the same test statistic, AMI, in a CRT helps mitigate this issue.

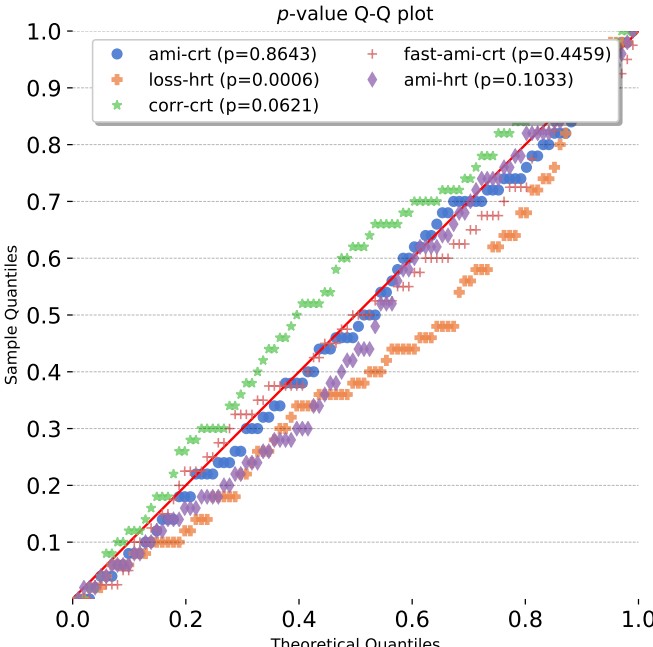

**Figure 3:** Quantile-Quantile plot showing uniformity of $p$-values across various FDR-controlling methods. A Kolmogorov-Smirnov (KS) test is performed to test if each set of $p$-values is uniformly distributed.

# E    CELIAC DISEASE GENOMIC FEATURE SELECTION

Table 6 shows the set of SNPs deemed significant by AMI-CRT. We annotate each SNP with its position in the human genome, and whether it was previously reported as significant in a biological study.

| Position | SNP | Featured in previous study |
|---|---|---|
| chr2:102454108 | rs917997 | yes |
| chr2:68371823 | rs17035378 | yes |
| chr3:159947262 | rs17810546 | yes |
| chr3:188394766 | rs1464510 | yes |
| chr3:46193709 | rs13098911 | yes |
| chr4:122194347 | rs13151961 | yes |
| chr6:137651931 | rs2327832 | yes |
| chr6:26451325 | rs2237236 | yes |
| chr6:28423688 | rs2859365 | no |
| chr6:29505139 | rs757256 | no |
| chr6:29844253 | rs2734994 | no |
| chr6:31642909 | rs1052486 | no |
| chr6:32638107 | rs2187668 | yes |
| chr6:90216893 | rs10806425 | yes |
| chr11:128511079 | rs11221332 | yes |
| chr12:111569952 | rs653178 | yes |
| chr21:44227538 | rs4819388 | yes |

**Table 6:** This table details each SNP returned by AMI-CRT, whether it was featured in a previous biological study relating to Celiac disease, and its position on the human genome.

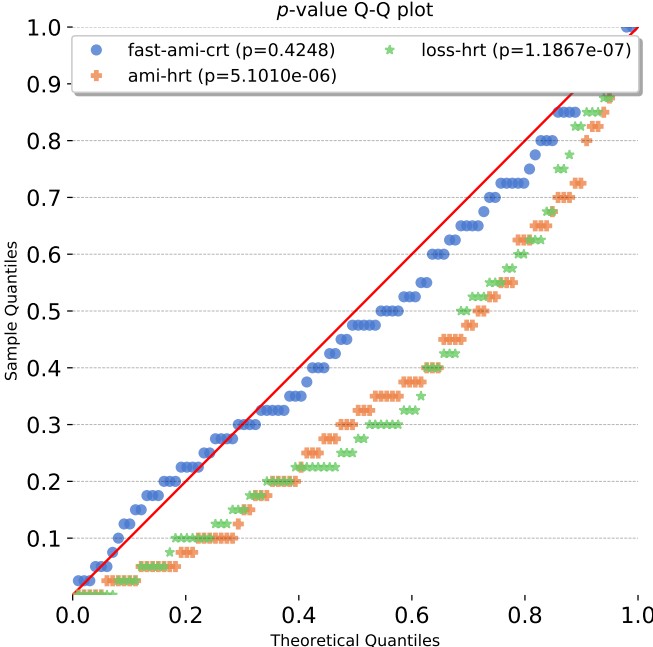

**Figure 4:** Quantile-Quantile plot showing uniformity of $p$-values across various FDR-controlling methods in the case of poor approximations of $q(\mathbf{x}_j \mid \mathbf{x}_{-j})$. A Kolmogorov-Smirnov (KS) test is performed to test if each set of $p$-values is uniformly distributed.

## F    HOSPITAL READMISSION FEATURE SELECTION

For the hospital readmission dataset, we applied several standard pre-processing techniques. We filtered each sample the data in a manner similar to (Strack et al., 2014):

- It is an inpatient encounter (a hospital admission).

- It is a diabetic encounter, that is, one during which any kind of diabetes was entered to the system as a diagnosis.

- The length of stay was at least 1 day and at most 14 days.

- Laboratory tests were performed during the encounter.

- Medications were administered during the encounter.

Further, we binarized the labels so that a label of $1$ indicates a readmission event, and a label of $0$ indicates no readmission event. We then encoded each categorical feature as a one-hot encoding. We then imputed missing values using the median across the dataset, and dropped the "weight" feature as it was found to be 97% missing.

To sample from the complete conditional distributions $q(\mathbf{x}_j \mid \mathbf{x}_{-j})$, we used a neural network to fit the complete conditional regression detailed in (Miscouridou et al., 2018). For continuous values of $\mathbf{x}_j$, we first discretized the data into bins, then used our neural network to predict the bins. To map the bins back to values in the domain of $\mathbf{x}_j$, we used the mean of the range of values in each bin.

## G    IMAGENET INSTANCE-WISE FEATURE SELECTION

Figure 5 shows some of the results of instance-wise feature selection on ImageNet data using AMI-CRT. Figures 6 and 7 show results on the same task, using LIME and SHAP respectively. We notice

that AMI-CRT identifies patches that seem more likely to help differentiate between ambulances and policevans. AMI-CRT identifies relevant text like the words "ambulance" or "police" that are very likely to help distinguish between the two classes. LIME identifies some relevant features of the image like wheels and lights, but fails to identify relevant words. SHAP does a good job at identifying distinguishing symbols like the caduceus and the FBI logo, but occasionally misses out on relevant text.

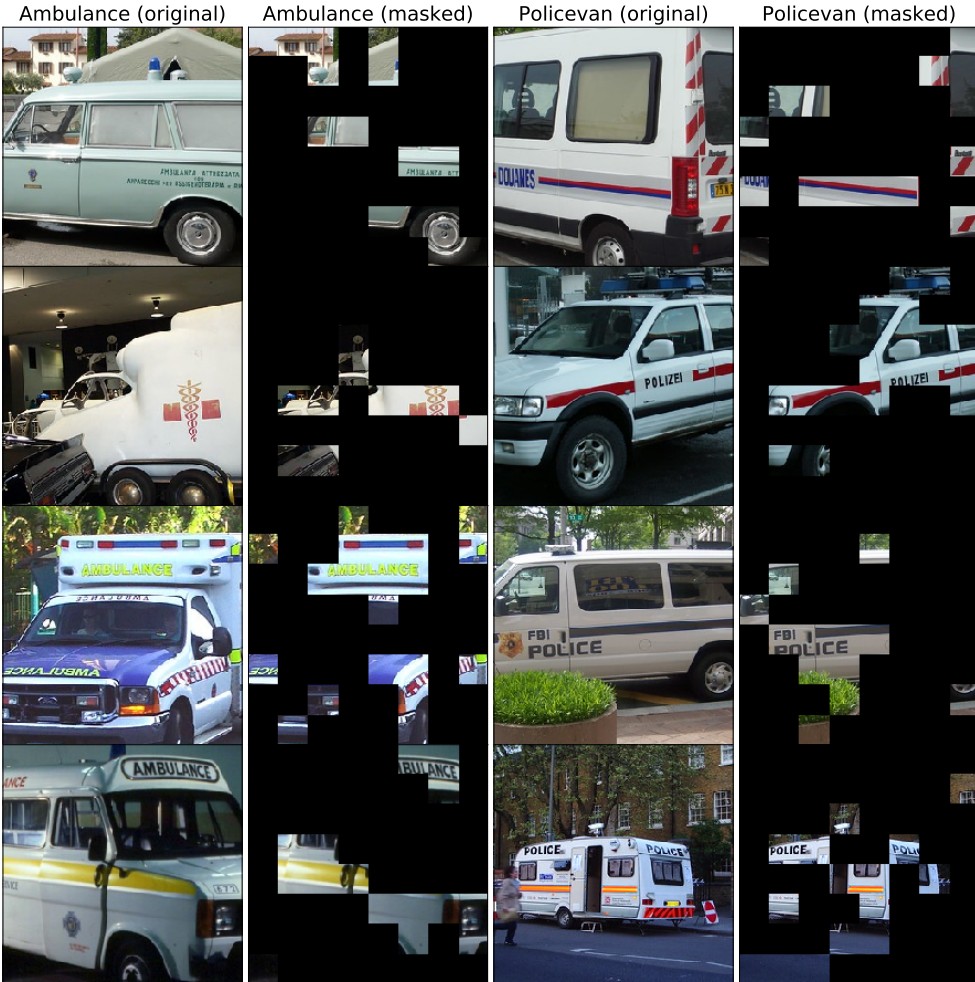

**Figure 5:** Instance-wise feature selection using AMI-CRT. The first and third columns show the original image of ambulances or policevans respectively. The second and fourth columns show only the patches which were found to have non-zero AMI with the label, given the rest of the patches.

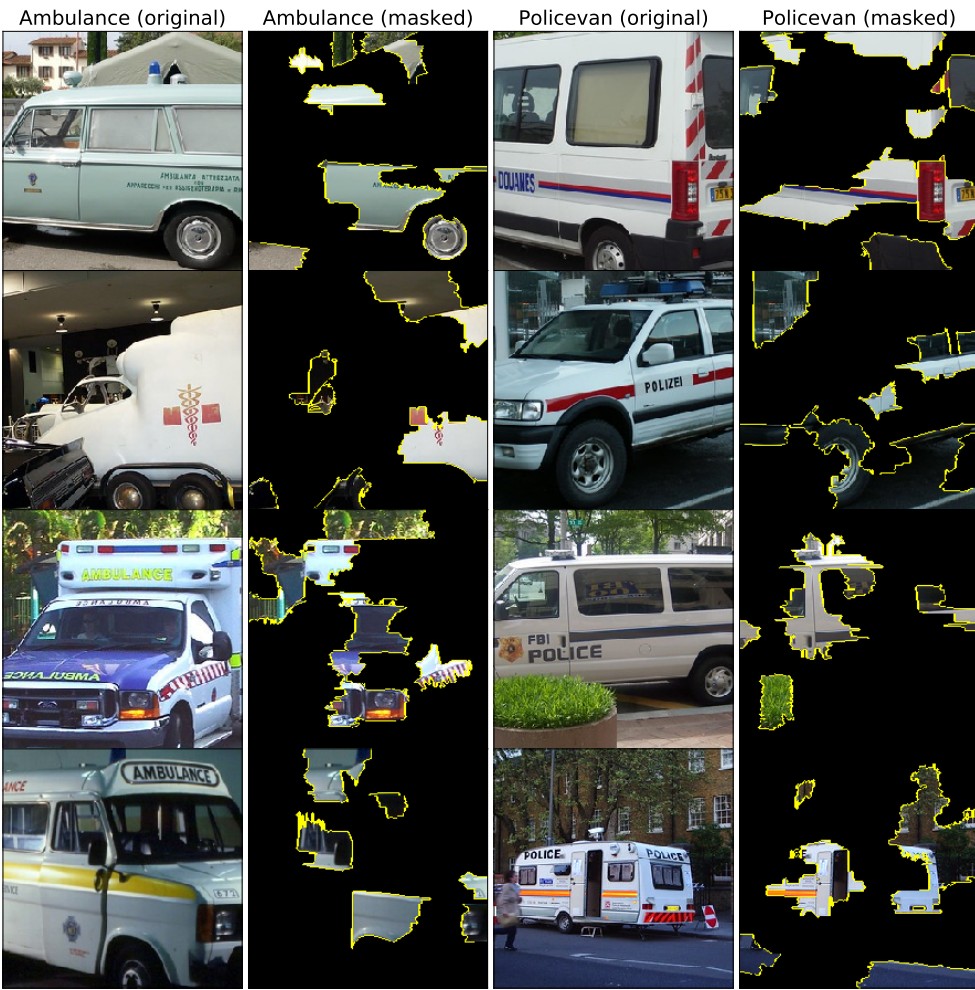

| Ambulance (original) | Ambulance (masked) | Policevan (original) | Policevan (masked) |

**Figure 6:** Instance-wise feature selection using LIME. The first and third columns show the original image of ambulances or policevans respectively. The second and fourth columns show only the patches which were found to be important.

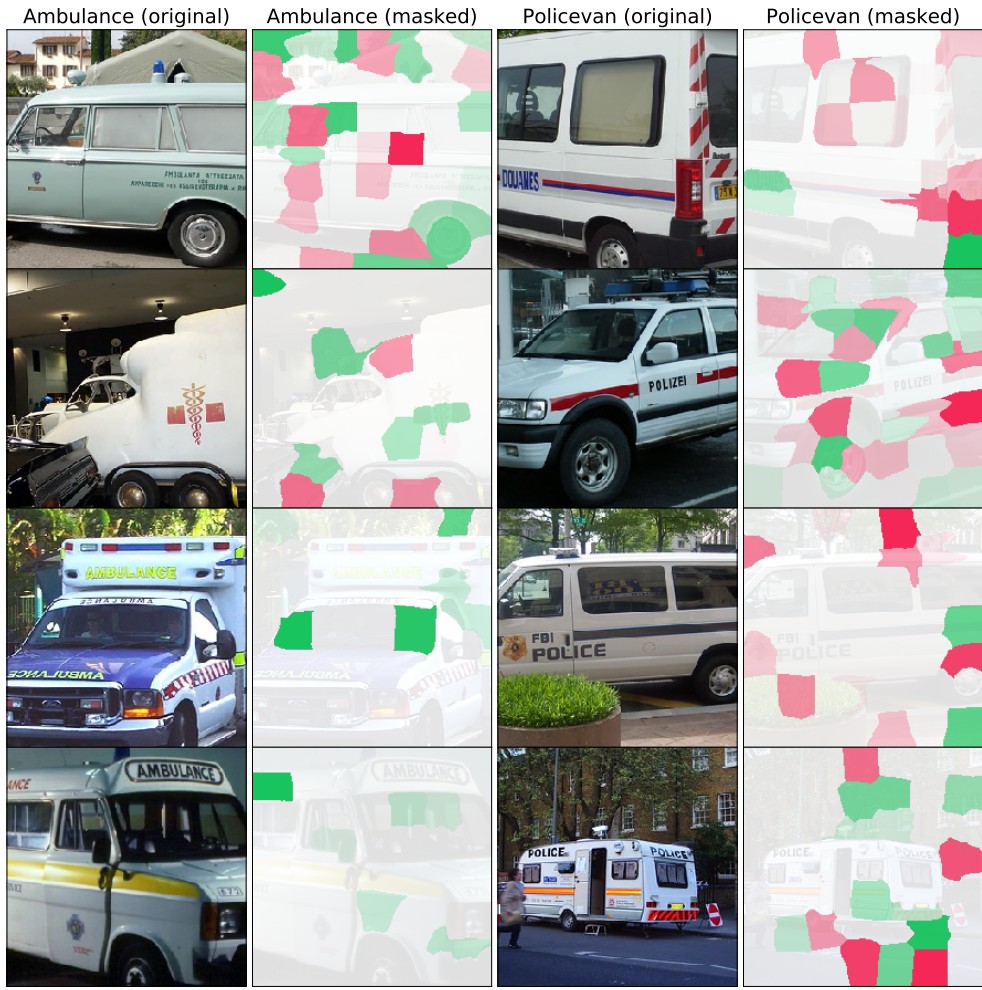

Ambulance (original)  Ambulance (masked)  Policevan (original)  Policevan (masked)

**Figure 7:** Instance-wise feature selection using SHAP. The first and third columns show the original image of ambulances or policevans respectively. The second and fourth columns show patches that are found to contribute to the label. Green indicates a patch found relevant for the ambulance class, and red indicates a patch found relevant for the policevan class.