# OpenReview forum: "Model-Agnostic Feature Selection with Additional Mutual Information"
_ICLR.cc/2020/Conference — Reject_

### Official Review · AnonReviewer1 · 2019-10-22
**Official Blind Review #1**

**Rating:** 6

**Review:**

This paper proposes a practical improvement of the conditional randomization test (CRT) of (Candes et al., 2018).
In the study of (Candes et al., 2018), the choice of the test statistic as well as how one estimates conditional distributions were kept open.
The authors proposed "proper test statistic" as a promising test statistic for CRT, and proved that f-divergence is one possible choice.
They further shown that KL-divergence has a nice property among possible f-divergences: KL-divergence cancels out some of the conditional distributions, and thus the users need to estimate only two conditional distributions to compute the test statistic.
For estimating those conditional distributions in the test statistic, the authors proposed fitting regression models.

Overall, I think the paper is well-written and the idea is stated clearly.
The use of KL-divergence for CRT seems to be reasonable.
The proposed algorithms look simple and easy to implement.

My only concern is on the practical applicability of the proposed algorithms (which, however, may be not a unique problem for this paper, but for all the CRT methods).
They require fitting regression models for each feature xj.
For high-dimensional data with more than thousands of features, fitting regression models for all the features seem to be impractical.
For the imagenet data experiment, the authors successfully avoided this problem by using an inpainting model.
However, this approach is apparently limited to image data.
I am interested in seeing if there is any promising way to make the algorithms scalable to high-dimensional data.


### Updated after author response ###
The authors have partially addressed my concern on the scalability of the proposed algorithm to high-dimensional data.
I therefore keep my score unchanged.

**Experience Assessment:**

I have read many papers in this area.

**Review Assessment: Checking Correctness Of Derivations And Theory:**

I assessed the sensibility of the derivations and theory.

**Review Assessment: Checking Correctness Of Experiments:**

I assessed the sensibility of the experiments.

**Review Assessment: Thoroughness In Paper Reading:**

I made a quick assessment of this paper.

---

> ### Author Response · Authors · 2019-11-13
> **Updates and clarifications regarding the paper**
>
> We thank the reviewer for their comments.
>
> *Computational expense*
> The reviewer points out correctly that fitting a model on each feature is computationally expensive in high-dimensional settings. While our algorithm is embarrassingly parallel, we added discussion about a faster version, the fast-ami-crt, which requires only 1 extra regression onto response per feature. While the fast version may not enjoy the theoretical guarantees of ami-crt, it proves effective empirically. We also explore the computational vs statistical tradeoffs in choosing between the regular and fast versions.
>
> In practice, knowledge about the data-generating process can be used. For example, in genetics, genes are independent of other genes located far away. Further, given groups of features, the inpainting method we use for images would apply to any data. We believe that automatically choosing such groups in a domain-agnostic manner is an interesting direction for the future extensions building on top of our methods.

---

### Official Review · AnonReviewer2 · 2019-10-22
**Official Blind Review #2**

**Rating:** 3

**Review:**

# Paper summary

This paper addresses supervised feature selection: given a D-dimensional input variable x = (x_1, ..., x_D), and a response variable y, the goal is to find a subset of "useful" features in x. Here, a feature x_j is useful if it is dependent on y even when conditioning on all other input variables (denoted by x_{-j}, which is a set). A generic procedure that can produce a p-value for each feature (allowing on to test each feature whether it is useful) is the conditional randomization test (CRT) proposed in Candes et al., 2018.  For the CRT to produce a valid p-value for each feature (input dimension) x_j, one needs to specify a test statistic that measures conditional dependence between x_j and y given the rest of the features.

This paper contributes the following results:

1. Propose using an estimate of the f-divergence for the conditional dependence measure and use it with the CRT (section 2.2).

2. Measuring the conditional dependence with an f-divergence requires estimating a few conditional density functions. The paper considers the KL divergence as a special of f-divergence. This particular choice turns out the reduce the number of conditional density functions that have to be estimated (section 2.3). The paper also shows that the resulting conditional measure coincides with what is known as the Additional Mutual Information (AMI) studied in Ranganath & Perotte, 2018.

3. The paper also studies instance-wise feature selection i.e., selecting a subset of input features which can explain the response specifically for one instance (example) x.  Yoon et al., 2019 proposed a criterion to decide the importance of a feature for instance-wise feature selection (definition 2). Briefly, a feature x_j is deemed important if q(y | full x) > q(y | x without the jth feature), where q is the conditional density function of y given x. _Contribution_: The paper notes that this criterion may fail and derive sufficient conditions (Definition 3) under which this approach will always work.

In simulation on toy problems, the paper shows that the proposed method (KL divergence + CRT) has the highest mean under the ROC curve (Table 2), compared to competing methods. In real problems on images, the paper shows that the proposed instance-wise feature selection can be used to select relevant image patches (features) that explain the class of the input images. The paper also conducts experiments on hospital readmission data (Section 4.3), and genomics data (Section 4.2).



# Review

The paper is overall well written with some parts that can be improved (details below). Introduction and related work in section 1 are easy to follow. The paper is also mostly self-contained and friendly to non-specialists who may not work on feature selection primarily. My concerns are

1. I find that the amount of contribution is not sufficient. CRT is known from Candes et al., 2018. The present paper proposes using KL-divergence with it. This can be interesting if the combination gives some clear advantages.  Unfortunately I do not find that this is the case. It turns out that one still needs to learn two conditional density functions (see lines 3-4 in Algorithm 1). Further and even more concerning, one has to refit another conditional density function *for each draw from the null distribution* (see "Fit regression" in the loop in Algorithm 1). As an intermediate step for solving the original feature selection problem, I find that learning conditional density functions is a much more difficult problem. All these limit the novelty of the idea. While the title of the paper contains "model-agnostic", the idea of fitting conditional density functions seems to contradict it. The paper could have considered some nonparametric conditional dependence measures but did not. For instance, see

Kernel-based Conditional Independence Test and Application in Causal Discovery
Kun Zhang, Jonas Peters, Dominik Janzing, Bernhard Schoelkopf
2012

and other papers that extend this paper.

Why was the approach of fitting conditional density functions chosen?

2. Related to the previous point, refitting a conditional density model for each draw from the null distribution must be very costly computationally. This point is never addressed in the paper.

3. Lemma 1 states that the expected f-divergence is a "proper statistic" (in the sense of Definition 1) i.e., p-value is uniformly distributed if the feature is not useful, and vanishes (asymptotically) if the feature is useful. This result unfortunately relies on a strong assumption that there is a consistent estimator for the f-divergence. In fact, the proof does not even rely on the fact that it is an f-divergence. It can be any divergence D(p,q) such that D(p,q) > 0 if  p!=q and D(p,p) = 0. In the proof in section C.2 in the appendix, existence of the quantile function $(F^{-1}_N)$ is never discussed. I can see the first part of the proof (under the alternative H1). But I do not see the second part (under H0). Since $\hat{f}$ is a consistent estimator by assumption, as N goes to infinity, the two quantities in the indicator function (in expectation) should both go to the same constant. Isn't this the case?

4. As a contribution, the paper states a sufficient condition in Definition 3 under which instance-wise feature selection with the approach in Definition 2 is *always* possible. When does the condition hold in practice? How do we know if it holds or not? If it does not, what can go wrong?

5. Toy experiments: What is D in Xor and Orange? Where is the "selector" problem in Table 2? In table 2, "lime" and "shap" also seem to perform well. The paper never explains why the proposed approach is better than other methods (only reporting higher mean are under the ROC curve). This should be possible for toy problems.



# Minor but does affect the evaluation

* Paragraph after Lemma 1: it is unclear why those conditional distributions are required instead of conditional distributions in Eq. 3.

* Section E.1 (appendix), page 16: I think you should have $N( 0.5x_1 + 0.5x_2, \sigma^2_\epsilon)$ instead of
$0.5N(x_1, \sigma^2_\epsilon) + 0.5N(x_2, \sigma^2_\epsilon)$.



# Things that can be improved. Did not affect the score.

* Section 1.1: the sentence about permutation tests is vague.

* Page 2, our contributions: "necessary" should be "sufficient"?

* Section 2, conditional randomization tests: This paragraph is unfortunately not well written even though it is a very important prerequisite of this work. For instance, at " ... replaced by samples of $\tilde{x}_j^{(i)}$ that is conditionally independent of the outcome...", at that point, it is unclear "conditioning on what". Following this sentence, one approach might be to replace $\tilde{x}_j^{(i)}$ with a constant (which is independent of everything else). It is not until definition 1 that this becomes clearer. Also, the "null hypothesis" (which is in the first line of equation 2) is never stated throughout the paper.

* Eq 1: that (i) is unclear. Should state that for i=1,...,N.

* Eq 2: rewrite the second line. The left hand side states that the p-value "converges in distribution to". The second line should be just 0.

* After eq.6, how to choose T (the number of bins) in practice?

* Definition 3 is actually a proposition? It is unclear what is being defined there.

* The word "complete conditional knockoffs (CCKs)" appears for the first time in Section 3.2 without any explanation.

* Orange skin on page 8: what is "~ exp(...)"? An exponential distribution, or just exponential function?


**Experience Assessment:**

I have published one or two papers in this area.

**Review Assessment: Checking Correctness Of Derivations And Theory:**

I carefully checked the derivations and theory.

**Review Assessment: Checking Correctness Of Experiments:**

I assessed the sensibility of the experiments.

**Review Assessment: Thoroughness In Paper Reading:**

I read the paper at least twice and used my best judgement in assessing the paper.

---

> ### Author Response · Authors · 2019-11-13
> **Updates and clarifications regarding the paper**
>
> We thank the reviewer for their comments.
>
> *Overall*
> We have made edits in response to this review and reviewer 1. Your summary helped us identify the necessary edits. We clarify our contributions here:
>
> We introduce the notion of proper test statistics. These are test statistics that, when used in a randomization test for feature selection, yield power that approaches 1 in the limit of data, and are uniformly distributed under the null hypothesis.
> We show estimators of expected divergences are proper test statistics. We develop AMI-CRT which uses the KL-divergence for a computational speedup over other divergences, and enables the reuse of model code for computing a null distribution. We also show that unlike the 0-1 loss, the log probability in the KL-divergence is smooth and therefore results better calibrated p-values.
> We develop sufficient conditions to perform instance-wise feature selection, show how our estimates can be adapted to this setting, and compare our methods to several state-of-the-art baselines on simulated and real datasets.
>
> *Proper-tests and AMI-CRT*
>
> In the original draft, we mention that the definition of a proper test statistic mirrors that of a proper scoring rule. So the advantages that proper test statistics offer are analogous to the advantages that proper scoring rules offer in supervised learning.
> Proper test statistics make minimal assumptions about the true data generating process and are asymptotically as powerful as any other test.
> We have since clarified the discussion to highlight the value of the KL divergence
> Using the KL-based statistic does not require computing  q(y | x_{-j}). This provides two important benefits: a) avoids learning from x and x_{-j} which have different dimensions and may require different model structures. For example, convolutional-networks require additional padding when learning from x_{-j}. This advantage also allows use to reuse our AMI-CRT framework to perform instance-wise feature selection. b) We compute one less conditional distribution per feature, thereby decreasing the amount of required computation.
>
> *Fitting regressions vs. conditional density models*
> We want to offer a clarification here. As discussed in the paper, feature selection involves testing conditional independence properties of the true data generating distribution in general. In this setting, knowledge of the true conditional density is required for feature selection.
> Moreover, the conditional density of interest is over a scalar response variable which we can solve via supervised learning. When supervised learning is hard, fitting a single model from the features to the response which all but the most basic methods require might be hard, thereby ruling out both prediction and selection.
>
>
> *Confusion regarding model-agnostic*
> We used the phrase 'Model-agnostic' to mean that our testing procedure does not depend on a specific model. Instead, we use whichever model fits best for a particular task.
>
> We have retitled our paper : ‘Black-box feature selection with Additional Mutual Information’ to resolve this confusion and clarify our contributions.
>
> *Refitting for each draw from the null*
> In the updated draft we include discussion about fast-ami-crt. Fast-ami-crt fits only a single null model for each feature (rather than one per draw from the null), and uses a mixture of the full model and the null model to compute the test-statistic. In our experiments, relative to baselines we show that the mixture model guards against errors poor quality samples from the null.
>
> *Lemma 1 details*
> The reviewer points out correctly that the proof works for any divergence that can detect equality of distribution. However, we suggest f-divergences are simply an example of proper test statistics. Therefore, our proof need not rely on f-divergences. Regardless, we updated the writing to make this fact more apparent.
>
> We note we only need the generalized inverse cumulative distribution function to guarantee uniformity of p-values. This exists when the cumulative distribution function of the test-statistic is continuous everywhere. We have clarified the discussion regarding this in the paper and updated the proof.
>
> Under the null hypothesis H0, for any sample size N, the distribution of the p-value will be uniform(0,1). The fact that the estimators converge to a single number does not have any implication on the convergence of the distribution of the p-value. The reason behind this is that the limit and comparison (indicator function) do not commute.

---

> > ### Comment · AnonReviewer2 · 2019-11-14
> > **Thanks for the update**
> >
> > Thanks for the updated version and your responses. The responses did clarify some questions I had. I still wonder about the motivation of the proposed approach (as opposed to, say, using some conditional dependence test statistic which does not require estimating conditional densities many times).

---

> ### Author Response · Authors · 2019-11-13
> **Updates and clarifications regarding the paper (Cont.)**
>
> *Instance-wise feature selection sufficient condition*
> In practice, this condition is achieved in tasks like image-recognition where the distribution y | x is sharp for most samples. This is reflected in the test error. As discussed in the instance-wise section, when perfect prediction does not hold, it may not be possible to distinguish between important and unimportant features for all samples. The original draft discusses this issue with an example in Section 3. In our experiments we have a noisy-selector, we show that while noise degrades performance across all methods, ami-iw still performs best.
>
>
> *Experiments*
> **Simulations**
> D is the dimension of the x variable. We include particular parameter choices for each experiment in the updated version. We have also added further discussion explaining the performance of our method. Briefly, the simulated examples demonstrate that while other methods do perform well with respect to certain metrics, only ami-crt satisfies all desiderata: high power, uniformity of p-values for null features, and control of False-discovery rate (captured by AUROC).
>
> *# Minor but does affect the evaluation*
>
> * conditional distributions after lemma 1 vs. conditional distributions in Eq. 3*.
>
> All f-divergences use the ratio of the conditional distributions in Eq. 3. This ratio simplifies to include the conditional distributions in the paragraph after lemma 1. We have noted that this might be hard to follow, and clarified this fact in the updated draft (final paragraph of section 2.1), rather than in the referenced appendix section.
>
> * section E.1 (appendix), Mixture of gaussians vs. gaussian density*
>
> There seems to be a misunderstanding here. In the original draft, section E.1 (appendix), page 16, N(mu, sigma) is a gaussian density, not a gaussian random variable. The average of two gaussian densities is not a gaussian density, rather a mixture of gaussians. (In the new draft, the derivation is moved to section C.1)
>
> *Things that can be improved. Did not affect the score.*
>
> * Section 1.1: the sentence about permutation tests is vague.*
>
> We have updated this sentence from “fail in the case of” to “fail to test conditional independence” for added clarity.
>
> * Page 2, our contributions: "necessary" should be "sufficient"?*
>
> We believe that proper-test statistics are necessary for feature selection without making further assumptions about the relationship between outcome and response.
>
> * Section 2, conditional randomization tests.*
>
> We have simplified the introduction to section 2 for clarity. The revised version better sets up the need for such tests in finite sample settings.
>
> * Eq 1: that (i) is unclear. Should state that for i=1,...,N.*
>
> This equation states that all samples in the dataset have their jth feature replaced by a sample from q(x_j | x_{-j}). The equation has been updated to further clarify this.
>
> * Eq 2: rewrite the second line. The left hand side states that the p-value "converges in distribution to". The second line should be just 0.*
>
> Equation 2 describes the convergence in distribution of the p-values. If the jth feature was not important, then this distribution is Uniform(0,1). Otherwise, it converges to a distribution where observing 0 has probability 1. We have clarified this in the draft.
>
> * After eq.6, how to choose T (the number of bins) in practice?*
>
> The choice of bins has been explored in the histogram approximation literature. We point the reviewer to the discussion in (Wasserman, 2006) and (Miscouridou, 2018).
>
> * Definition 3 is actually a proposition? It is unclear what is being defined there.*
>
> Definition 3 has been updated to Proposition 1.
>
> * The word "complete conditional knockoffs (CCKs)" appears for the first time in Section 3.2 without any explanation.*
>
> We have removed the terminology CCK and instead refer to the object q( x_ j | x_{ -j } ) itself as necessary.
>
> * Orange skin on page 8: what is "~ exp(...)"? An exponential distribution, or just exponential function?*
>
> The discussion in the experiments about the data generation processes has been clarified.
>
> We have updated the paper to clarify all definitions and added required information.

---

### Official Review · AnonReviewer3 · 2019-10-24
**Official Blind Review #3**

**Rating:** 3

**Review:**

This paper presents a method to provide some level of interpretation on the influence of input features on the response of a machine level model all the way down to the instance level. The proposed method is model agnostic. Quoting the authors, they advocate for methods that look at interpretability “as understanding the population distribution through the lens of the model” without restriction on the models fit. The problem is posed as a hypothesis testing problem. The paper proposes “proper test statistics” for model agnostic feature selection. It is argued that f-divergence tests are proper statistic tests, with the KL being particularly interesting as it provides computational advantages.

I have found the paper interesting. The topic is relevant and the approach is interesting. However, I have two main reservations for this work. First, I have found the method difficult to follow and sometimes unclear. Important results are only explained in the appendix. For instance, the derivation of Equation 5 is important but only shown in the appendix. Furthermore, that derivation in the appendix needs to be clarified in my view. For instance, on page 15, for the derivation of $\delta_I$, can you explain how you went from the second equality to the third equality where references to \tilde{x}_j are removed from one line to another? It could be due to your definition for the term with a conditional independence	with the outcome assumed but I suggest that you clarify this as it is important for the paper and for the use of the KL. Also, in this equation, should it be $q(x_j|x_{-j}) instead of $q(x_j,x_{-j})$?

The second issue that I have is with the experiments. Any reason why the key results on the interpretability of the approach are mostly shown in the appendix (e.g., table 4,5,6)?  Why does table 6 not show results for all the baselines? For the hospital readmission use case, were you able to also get percentages of important features and have it compared with the baselines and vetted for clinical significance? This is more minor but worth double checking in my opinion. For this experiment on re-admission, the paper claims to have data from 130 hospitals for 10 years. Yet the n numbers seems pretty small to me. Total number of events < 100 000 for 130 institutions over 10 years. That would mean that we are dealing with less than an average of 80 admissions per institutions per year. Please confirm or explain if any filtering was done beyond what is described in appendix I.

**Experience Assessment:**

I have read many papers in this area.

**Review Assessment: Checking Correctness Of Derivations And Theory:**

I assessed the sensibility of the derivations and theory.

**Review Assessment: Checking Correctness Of Experiments:**

I assessed the sensibility of the experiments.

**Review Assessment: Thoroughness In Paper Reading:**

I read the paper at least twice and used my best judgement in assessing the paper.

---

> ### Author Response · Authors · 2019-11-13
> **Updates and clarifications regarding the paper**
>
> We thank the reviewer for their comments.
>
> *Proper tests and AMI-CRT*
>
> Thank you for the comment about divergences. We have updated the discussion about divergences. It now includes the derivation of delta_j and highlights the advantages of the KL divergence. See section 2.2 for the full derivation.
>
> We added explanations for each step in the derivation for delta_j. \tilde{x}_j is sampled anew from x_j | x_{-j}. Any dependence between y and \tilde{x}_j is therefore broken by conditioning on x_{-j}.
>
> *Experiments*
>
> In response to the review, we have reorganized the paper to include the presentation of these important results in the main text.
>
> *FDR controlling methods*
>
> The Benjamini-Hochberg correction (1995) requires p-values to control the FDR. This table shows only those methods that produce p-values so that FDR can be controlled. We have made this fact more explicit in the main text.
>
> *Hospital readmission *
>
> Yes, we reference a citation (Strack et. al. 2014) that clinically validates features from the hospital readmissions dataset and selects a subset as important. These features are used to compute our ROC curves.
> The dataset is from Strack et al (2014). Here is the filtering criteria they apply:
> (1)	It is an inpatient encounter (a hospital admission).
> (2)	It is a diabetic encounter, that is, one during which any kind of diabetes was entered to the system as a diagnosis.
> (3)	The length of stay was at least 1 day and at most 14 days.
> (4)	Laboratory tests were performed during the encounter.
> (5)	Medications were administered during the encounter.
> No additional samples were dropped from this dataset in our experiments. These steps have been added to the appendix.

---

### Decision · Program_Chairs · 2019-12-19

**Decision:**

Reject

**Comment:**

The paper presents an approach to feature selection. Reviews were mixed and questions whether the paper has enough substance, novelty, the correctness of the theoretical contributions, experimental details, as well as whether the paper compares to the relevant literature.